# Right inferior frontal gyrus damage is associated with impaired initiation of inhibitory control, but not its implementation

Yoojeong Choo[1,2], Dora Matzke[3], Mark D Bowren Jr[4], Daniel Tranel[1,5], Jan R Wessel[1,2,5]*

[1]Department of Psychological and Brain Sciences, University of Iowa, Iowa City, United States; [2]Cognitive Control Collaborative, University of Iowa, Iowa City, United States; [3]Department of Psychology, University of Amsterdam, Amsterdam, Netherlands; [4]Department of Clinical and Health Psychology, University of Florida, Gainesville, United States; [5]Department of Neurology, University of Iowa Hospitals and Clinics, Iowa City, United States

*For correspondence:
jan-wessel@uiowa.edu

Competing interest: The authors declare that no competing interests exist.

**Abstract** Inhibitory control is one of the most important control functions in the human brain. Much of our understanding of its neural basis comes from seminal work showing that lesions to the right inferior frontal gyrus (rIFG) increase stop-signal reaction time (SSRT), a latent variable that expresses the speed of inhibitory control. However, recent work has identified substantial limitations of the SSRT method. Notably, SSRT is confounded by trigger failures: stop-signal trials in which inhibitory control was never initiated. Such trials inflate SSRT, but are typically indicative of attentional, rather than inhibitory deficits. Here, we used hierarchical Bayesian modeling to identify stop-signal trigger failures in human rIFG lesion patients, non-rIFG lesion patients, and healthy comparisons. Furthermore, we measured scalp-EEG to detect β-bursts, a neurophysiological index of inhibitory control. rIFG lesion patients showed a more than fivefold increase in trigger failure trials and did not exhibit the typical increase of stop-related frontal β-bursts. However, on trials in which such β-bursts did occur, rIFG patients showed the typical subsequent upregulation of β over sensorimotor areas, indicating that their ability to implement inhibitory control, once triggered, remains intact. These findings suggest that the role of rIFG in inhibitory control has to be fundamentally reinterpreted.

## Editor's evaluation

This study takes a fresh view of the hypothesis that right inferior frontal gyrus is critical in inhibitory control in humans, as assessed by the widely used stop-signal task. It applies recent development in modeling and EEG measures in patients with focal brain damage, yielding causal insights. The findings are important and the evidence is convincing.

## Introduction

Humans have remarkable cognitive control abilities, which allow them to safely navigate complex everyday situations. For example, when crossing a street, humans can rapidly stop themselves from continuing to walk when they suddenly notice a rapidly approaching car. The process underlying this ability to stop an already-initiated action is inhibitory control. In the laboratory, inhibitory control is

typically tested in the stop-signal task (*Verbruggen et al., 2019*), in which an initial go-signal (i.e., a cue to initiate a movement) is sometimes followed by a subsequent stop-signal, prompting the cancellation of that movement. The processes underlying behavior in the stop-signal task are described in a well-characterized cognitive model – the horse-race model (*Logan and Cowan, 1984*). This model purports that on each trial, the inhibitory process triggered by the stop-signal races with the movement-initiation process triggered by the go-signal, thereby determining whether an action can be successfully stopped. The assumptions of the horse-race model allow the calculation of stop-signal reaction time (SSRT) – a latent variable that expresses the speed of the stopping process, which is not otherwise overtly observable (as successful stopping is defined by the absence of a response). In a seminal study on the neural basis of inhibitory control, *Aron et al., 2003* showed that lesions to the right inferior frontal gyrus (rIFG) are associated with an elongation of SSRT, prompting the proposal that 'response inhibition can be localized to a discrete region of the PFC,' namely, the rIFG. This finding has spawned a wider, highly influential theory of inhibitory control, which, in its most recent iteration, holds that 'rIFG implements a brake over response tendencies' (*Aron et al., 2014*). While some subsequent lesion studies have cast some doubt upon specific claims of this theory (e.g., by demonstrating that other regions outside of rIFG lead to comparable deficits, e.g., *Picton et al., 2007*; *Swick et al., 2008*; *Yeung et al., 2021*) or have reported conflicting findings (*Floden and Stuss, 2006*), the crucial role of rIFG in action-stopping, indicated by the increase in SSRT in rIFC lesion populations, is still widely prevalent in current-day neuroscientific theory.

However, the SSRT method underlying this (and other) seminal work has recently undergone several substantial challenges (e.g., *Bissett et al., 2021*; *Matzke et al., 2017b*). One of the most prominent shortcomings of SSRT is that it does not account for trigger failures (*Band et al., 2003*) – trials with stop-signals in which inhibitory control process was never initiated to begin with. On such trials, erroneously executed responses do not result from an insufficiently fast stop-process losing the horse race, but from the mere fact that inhibitory control never 'entered the race.' Crucially, including such trials in the SSRT calculation leads to artificially inflated SSRT estimates and – in the worst case – can produce fictitious group differences in inhibitory control speed, which are instead more likely due to attentional lapses (e.g., *Matzke et al., 2017a*). In other words, in the abovementioned example of crossing a street, a trigger failure could be indicative of a deficit in noticing the approaching car, rather than in intercepting the walking movement fast enough.

In light of this confound, it is necessary to reassess the causal link between stop-signal performance and rIFG damage. Therefore, we here repeated the original lesion study by *Aron et al., 2003* with a stop-signal task that was optimized to implement a hierarchical Bayesian technique for the identification of trigger failures (*Matzke et al., 2017b*). We specifically tested whether rIFG lesion patients show increased trigger failure rates, especially in light of the fact that rIFG is a key region that is often implicated in stimulus-driven attention more generally (*Corbetta and Shulman, 2002*). Moreover, we extended the behavioral investigation by Aron and colleagues by measuring scalp-EEG. Specifically, we aimed to investigate the influence of rIFG lesions on stop-related β-bursts dynamics. β-Bursts are a recently discovered neurophysiological signature of inhibitory control (*Diesburg et al., 2021*; *Wessel, 2020*) and can provide additional insights into the distinction between the initiation and the implementation of inhibitory control. Specifically, β-burst rates over frontal cortex are increased on stop-compared to go-trials (*Enz et al., 2021*; *Wessel, 2020*; *Jana et al., 2020*), which has been proposed to reflect the initial stage of the inhibitory control cascade that ultimately results in action-stopping. In other words, frontal β-bursts purportedly reflect the *initiation* of inhibitory control. At the other end of the cascade, β-bursts over sensorimotor areas can be used to measure the successful *implementation* of inhibitory control. Sensorimotor β activity reflects an inhibited state of the motor system at baseline (*Kilavik et al., 2013*; *Soh et al., 2021*), and stop-related frontal β-bursts are followed by a rapid re-instantiation of these sensorimotor bursts (*Wessel, 2020*). This purportedly reflects the successful implementation of the inhibitory cascade and a return to an inhibited motor system (*Diesburg et al., 2021*).

Hence, in line with our behavioral hypothesis, we predicted that if rIFG lesion patients showed increased trigger failure rates, they would also show reduced frontal β-burst rates compared to both non-rIFG patients and healthy adult comparisons, reflecting a deficit in triggering/initiating inhibitory control. However, we furthermore predicted that when a frontal β-burst does take place in rIFG legion patients (i.e., when the cascade is successfully triggered), sensorimotor β-burst rates would

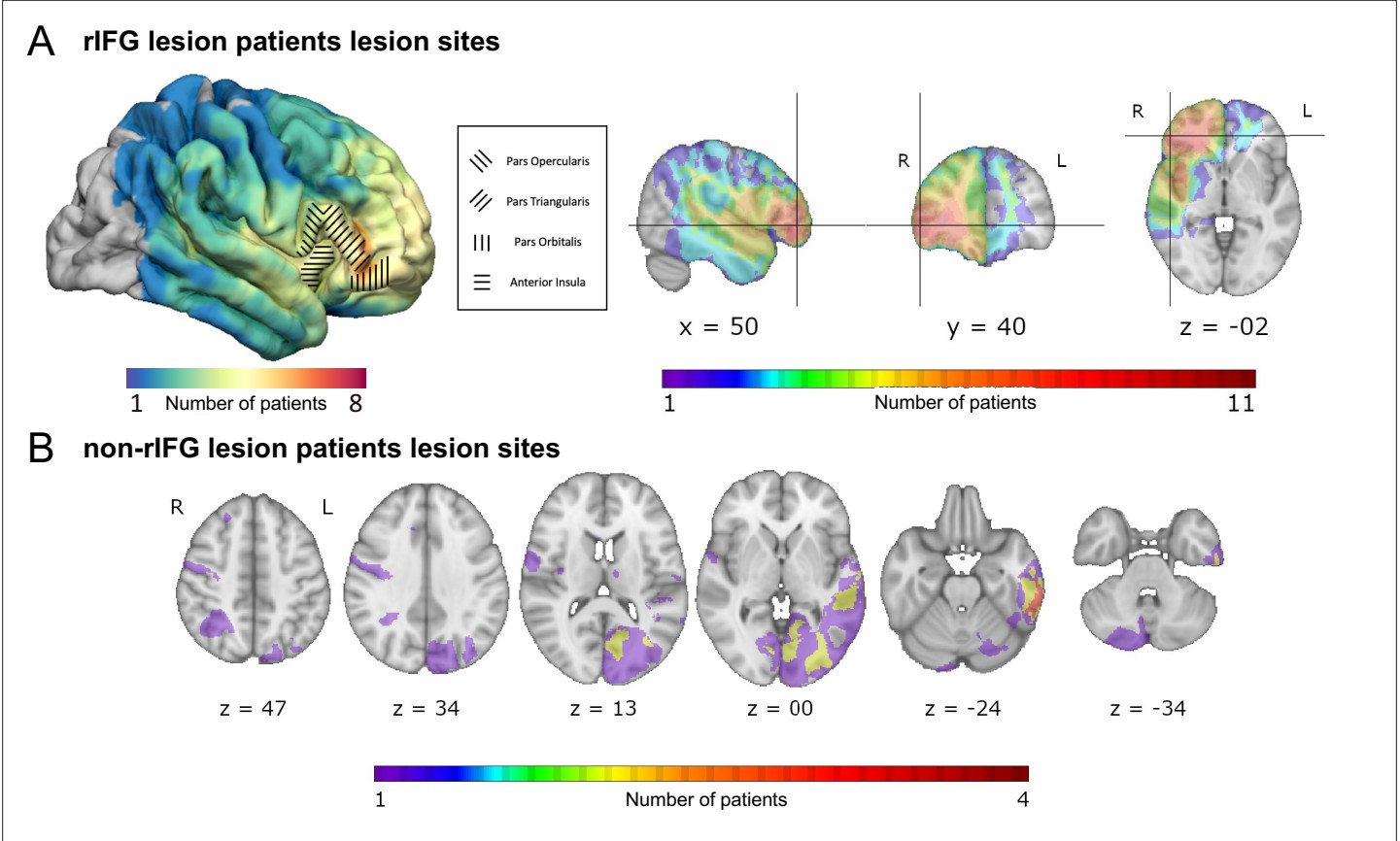

**Figure 1.** Overlap of lesions in patients with right inferior frontal gyrus (rIFG) lesions (rIFG lesion patients, top, **A**) and in patients with lesions outside of rIFG (non-rIFG lesion patients, bottom, **B**). (**A**) Top left: a lateral view of the lesion overlaps for rIFG lesion patients at the cortical level. Within the rIFG, three subregions are labeled: pars opercularis, pars triangularis, and pars orbitalis. The anterior insula is also highlighted. Top right (in order): sagittal, coronal, and axial views of the lesion overlap for rIFG lesion patients. The crosshair is centered on the right inferior frontal gyrus (IFG) on the Harvard-Oxford Atlas. All rIFG patients were included (N = 16). (**B**) Bottom: axial view of the lesion overlap for non-rIFG lesion patients. One lesion mask for a patient was missing and not included (N = 15). The color bar indicates the number of patients overlapped in lesion sites. R: right; L: left.

be appropriately upregulated, reflecting a retained ability to implement inhibitory control, despite damage to rIFG.

## Results

### Participants

Participants included 16 rIFG lesion patients, 16 non-rIFG lesion patients, and 32 age- and sex-matched comparisons. Lesion overlap maps are provided in *Figure 1*, and demographic data for

**Table 1.** Demographic information all four groups.

| Group | Sex | Handedness | Age | Chronicity |
|---|---|---|---|---|
| rIFG lesions | 10M/6F | 16R/0L | 54.25 (15.19) | 19.88 (21.54) |
| rIFG comparison | 10M/6F | 15R/1L | 54.63 (15.28) | n/a |
| Non-rIFG lesions | 10M/6F | 13R/3L | 61.50 (14.62) | 10.97 (15.97) |
| Non-rIFG comparison | 10M/6F | 15R/1L | 61.94 (14.40) | n/a |

M = male. F = female. R = right-handed. L = left-handed; rIFG = right inferior frontal gyrus. Age: mean age at testing in years (standard deviation); Chronicity: median length of time between lesion onset and current experiment in years (inter-quartile range).

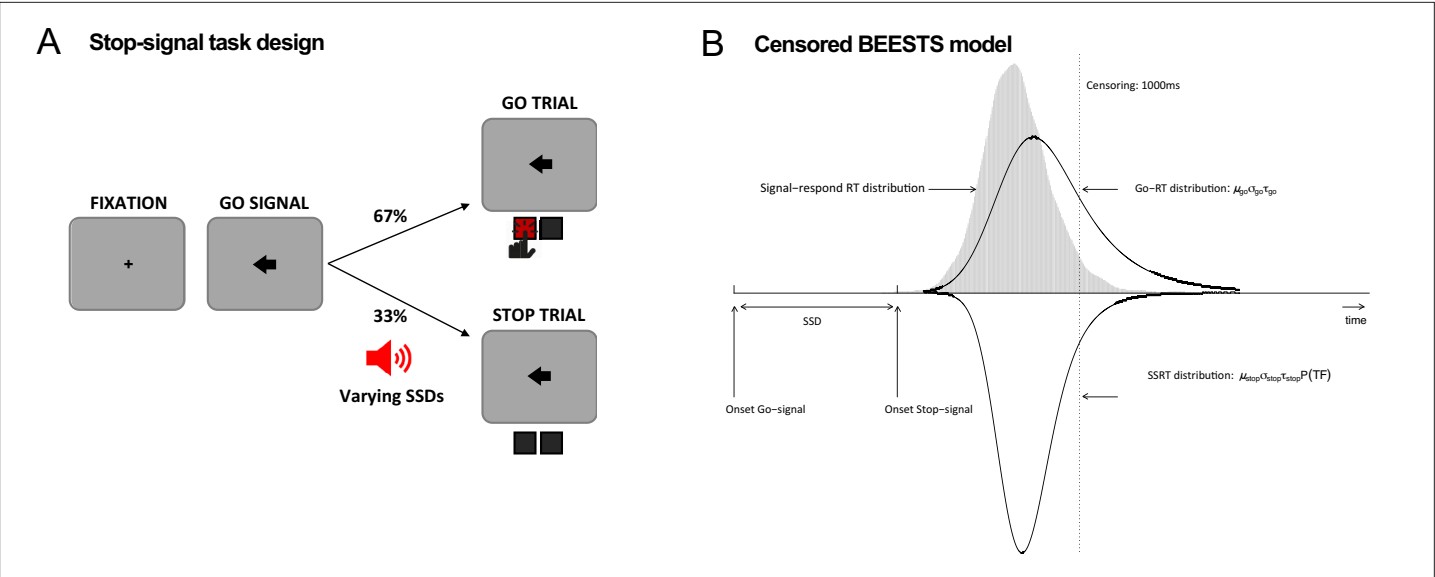

**Figure 2.** Task diagram and BEESTS model. (**A**) Schematic design of the stop-signal task. (**B**) Graphical representation of the censored BEESTS model. The model assumes that the finishing times of the go (Go-RT distribution) and the stop runner (stop-signal reaction time [SSRT] distribution) follow an ex-Gaussian distribution with parameters $\mu_{go}$, $\sigma_{go}$, and $\tau_{go}$, and $\mu_{stop}$, $\sigma_{stop}$, and $\tau_{stop}$, respectively. The finishing time distributions, and hence also the observed distribution of signal-respond RTs, are assumed to be censored above at 1000 ms to accommodate trials where the runners did not finish before the response window. The trigger failure parameter, P(TF), quantifies the probability that the stop runner was not initiated.

all participants are presented in *Table 1*. All participants performed a version of the stop-signal task (*Figure 2A*) that was optimized for the usage of BEESTS, a hierarchical Bayesian modeling technique that simultaneously accounts for the shape of Go-RT and SSRT distributions and the prevalence of trigger failures in the stop-signal task (*Matzke et al., 2013*; *Matzke et al., 2017b*, *Figure 2B*). This was a typical version of the stop-signal task (*Verbruggen et al., 2019*), which, however, included more trials than necessary for standard analyses of SSRT (see Task and Procedure).

## Behavioral and modeling results

BEESTS assumes that the finishing times of the go (Go-RT distribution) and the stop runners (SSRT distribution) follow an ex-Gaussian distribution with parameters **μ**, **σ**, and **τ**. The **μ** and **σ** parameters reflect the mean and the standard deviation of the Gaussian component and **τ** gives the mean of the exponential component and reflects the slow tail of the distribution. The mean and variance of the finishing time distributions can be obtained as **μ + τ** (i.e., mean Go-RT and SSRT) and $\sigma^2 + \tau^2$, respectively. Using a mixture-likelihood approach, the model can be augmented with a parameter, P(*TF*), that quantifies the probability that participants fail to trigger the stop runner (*Matzke et al., 2017b*). *Table 2* presents the posterior means and the corresponding 95% CIs of the population-level mean parameters in the four groups. The overlap of the posterior distributions of the go parameters indicated that $\mu_{go}$ was lower, whereas $\tau_{go}$ was higher in the rIFG lesion group relative to matched comparisons, suggesting that the Go-RT distribution of the lesion group had a faster leading edge but a larger skew. Resulting from the nearly perfect trade-off between $\mu_{go}$ and $\tau_{go}$, mean Go-RT did not differ between the two groups. The overlap of the posterior distributions of the stop parameters indicated that $\mu_{stop}$, $\sigma_{stop}$, and mean SSRT were higher in the rIFG lesion group than in matched comparisons, although the difference in $\sigma_{stop}$ was small. Crucially, we found a more than fivefold increase in the P(TF) parameter of the lesion group (16% vs. 3% in healthy comparisons). These results suggest that poor stop-signal performance associated with rIFG lesions is mainly attributable to increased trigger failure rate and a slowing of SSRT as a result of a shift in the leading edge of the distribution. Following *Matzke et al., 2017b*, we interpret this pattern of SSRT differences (increased $\mu_{stop}$ and constant $\tau_{stop}$) to reflect differences in the speed of encoding the stop-signal and not differences in the decisional or inhibitory component of SSRT, suggesting an attentional deficit in rIFG lesion patients. In contrast, with the exception of $\sigma_{stop}$, our analyses did not indicate the presence of differences in go or stop

**Table 2.** Posterior means and 95% credible intervals (CIs) of the population-level mean parameters in the four groups.

| | | Lesion patients | | Matched comparisons | | Difference | | Bayesian p |
|---|---|---|---|---|---|---|---|---|
| | | Posterior mean | 95% CI | Posterior mean | 95% CI | Posterior mean | 95% CI | |
| rIFG | $\mu_{go}$ | 479 | [435,521] | 578 | [505,650] | −99 | [−181,−15] | **0.01** |
| | $\sigma_{go}$ | 75 | [60,94] | 90 | [73,107] | −15 | [−39,10] | 0.12 |
| | $\tau_{go}$ | 154 | [111,216] | 76 | [54,109] | 78 | [25,145] | **0** |
| | Mean Go-RT | 633 | [572,708] | 654 | [576,730] | −21 | [−124,80] | 0.34 |
| | $\mu_{stop}$ | 223 | [197,249] | 198 | [182,215] | 25 | [−6,55] | 0.06 |
| | $\sigma_{stop}$ | 40 | [26,58] | 24 | [16,37] | 16 | [−3,37] | 0.05 |
| | $\tau_{stop}$ | 45 | [30,65] | 34 | [22,50] | 11 | [−12,35] | 0.17 |
| | Mean SSRT | 268 | [237,298] | 232 | [211,254] | 35 | [−2,73] | 0.03 |
| | P(TF) | 0.16 | [0.08,0.29] | 0.03 | [0,0.12] | 0.13 | [0.01,0.27] | **0.02** |
| Non-rIFG | $\mu_{go}$ | 547 | [504,589] | 575 | [506,646] | −28 | [−109,52] | 0.24 |
| | $\sigma_{go}$ | 98 | [84,111] | 104 | [85,122] | −6 | [−29,17] | 0.30 |
| | $\tau_{go}$ | 106 | [79,139] | 85 | [61,116] | 21 | [−19,62] | 0.15 |
| | Mean Go-RT | 653 | [602,706] | 660 | [586,735] | −7 | [−97,83] | 0.43 |
| | $\mu_{stop}$ | 215 | [191,239] | 199 | [184,215] | 16 | [−11,43] | 0.12 |
| | $\sigma_{stop}$ | 54 | [24,104] | 24 | [14,36] | 31 | [0.11,80] | **0.02** |
| | $\tau_{stop}$ | 29 | [18,44] | 26 | [13,44] | 3 | [−18,22] | 0.36 |
| | Mean SSRT | 244 | [218,271] | 225 | [204,249] | 19 | [−12,49] | 0.11 |
| | P(TF) | 0.02 | [0,0.05] | 0.04 | [0.02,0.08] | −0.02 | [0.07,0.02] | 0.09 |

The parameters of the Go-RT and SSRT distributions are presented on the ms scale. The P(TF) parameter is presented on the probability scale. The posterior distribution of 'Difference' is computed by subtracting the posterior samples of the matched comparison group from the corresponding samples of the lesion group, i.e., positive values indicate that the parameter is higher in the lesion group than in the matched comparison group. 'Bayesian p' is computed as the proportion of posterior samples in the posterior distribution of the lesion group that is larger than in the matched comparisons group. Bayesian p values are presented as P = min(p, 1p), i.e., as non-directional tests, with results at a two-sided p of 0.05 (i.e., a one-sided critical p of 0.025) highlighted in bold.

rIFG = right inferior frontal gyrus; SSRT = stop-signal reaction time.

**Table 3.** Posterior mean and 95% credible interval (CI) of the population-level mean difference between the two lesion groups (rIFG lesion vs. non-rIFG lesion).

| | Posterior mean of difference | 95% CI of difference | Bayesian p |
|---|---|---|---|
| $\mu_{go}$ | –68 | [-126,–10] | **0.02** |
| $\sigma_{go}$ | –22 | [–44,1] | **0.02** |
| $\tau_{go}$ | 49 | [–6,116] | 0.04 |
| Mean Go-RT | –19 | [–103,71] | 0.32 |
| $\mu_{stop}$ | 8 | [–26,44] | 0.32 |
| $\sigma_{stop}$ | –15 | [–67,24] | 0.33 |
| $\tau_{stop}$ | 16 | [–5,39] | 0.07 |
| Mean SSRT | 24 | [–18,63] | 0.12 |
| P(TF) | 0.15 | [0.06,0.28] | **0** |

The parameters of the Go-RT and SSRT distributions are presented on the ms scale. The P(TF) parameter is presented on the probability scale. The posterior distribution of 'Difference' was computed by subtracting the posterior samples of the non-rIFG group from the corresponding samples of the rIFG lesion group; i.e., positive values indicate that the parameter is higher in the rIFG lesion group than in the non-rIFG lesion group. 'Bayesian p' is computed as the proportion of posterior samples in the posterior distribution of the rIFG lesion group that was larger than in the non-rIFG lesion group. Bayesian p values are presented as $P = \min(p, 1p)$, i.e., as non-directional tests, with results at a two-sided p of 0.05 (i.e., a one-sided critical p of 0.025) highlighted in bold.
rIFG = right inferior frontal gyrus; SSRT = stop-signal reaction time.

parameters between non-rIFG lesion patients and their matched comparisons, neither did we find evidence for a difference in the probability of trigger failures. We also directly compared the two lesion groups. *Table 3* presents the posterior mean and the corresponding 95% CIs of the population-level mean difference between the two lesion groups. The overlap of the posterior distributions of the go parameters indicated that $\mu_{go}$ and the $\sigma_{go}$ were lower in the rIFG lesion group compared to the non-rIFG lesion group, whereas $\tau_{go}$ was higher. Crucially, the difference in the P(TF) parameter was large and reliable as the rIFG lesion group showed high trigger failure rate while the non-rIFG lesion group did not (16% vs. 2% in the non-rIFG lesion group). These patterns were similar to the comparison between the rIFG lesion group and their matched healthy comparison group. Taken together, our results are mostly consistent with an attentional rather than inhibitory account of stop-signal deficits associated with rIFG lesions.

Observed behavioral metrics of stop-signal performance, including non-parametric SSRT estimates computed with the integration method with replacement of go omissions (*Verbruggen et al., 2019*), can be found in *Table 4*. Compared to BEESTS, the integration approach estimated elongated SSRT in three groups except the healthy comparison for the rIFG lesion group; rIFG lesion: –39 ms, non-rIFG lesion: –11 ms, healthy comparison for rIFG lesion: 1 ms, and healthy comparison for non-rIFG lesion: –13 ms. Among them, the rIFG lesion group showed the largest attenuation. This suggests that the high trigger failure rate in the rIFG lesion group indeed inflated non-parametric SSRT (e.g., *Doekemeijer et al., 2021*; *Matzke et al., 2017a*; *Skippen et al., 2019*). Statistical comparisons for non-parametric SSRT, P(Go error), and P(Go miss) between groups can be found in Appendix 1.

## EEG results: Frontal β-bursts

*Figure 3* shows the results of the analysis of frontal β-bursts. The 2 × 3 ANOVA of frontal β-burst rates for the rIFG group and their matched comparisons revealed a main effect of TRIAL TYPE ($F_{(2, 60)}$ = 3.749, p=0.029, $\eta^2_P$ = 0.111) and, crucially, a significant interaction between GROUP and TRIAL TYPE ($F_{(2, 60)}$ = 4.215, p=0.019, $\eta^2_P$ = 0.123). The main effect of GROUP was not significant ($F_{(1, 30)}$ = 0.046, p=0.832, $\eta^2_P$ = 0.002).

Planned contrasts via paired-samples *t*-tests revealed that the significant interaction was specifically due to a significant reduction of the Successful-stop vs. Go difference in the rIFG lesion group (*M* = –0.005, SD = 0.116) compared to the matched healthy comparison group (*M* = 0.126, SD = 0.148), (*t*(15) = –3.862, p = 0.002, *d* = –0.965), with a large effect size. The Failed-stop vs. Go difference were also numerically reduced in the rIFG lesion group (*M* = –0.018, SD = 0.106) compared to matched healthy comparisons (*M* = 0.054, SD = 0.150), though that difference was not significant, (*t*(15) = –1.720, p=0.106, *d* = –.430).

**Table 4.** Non-parametric integration estimates and behavioral metrics of stop-signal performance in the four groups.

| | | Lesion patients | Matched comparisons |
|---|---|---|---|
| | | Mean (SD) | Mean (SD) |
| rIFG | Mean SSD | 286 (120) | 397 (131) |
| | Go RT | 589 (88) | 634 (108) |
| | SSRT | 307 (124) | 231 (43) |
| | SR-RT | 526 (76) | 569 (109) |
| | P(Go miss) | 0.058 (0.083) | 0.029 (0.04) |
| | P(Go error) | 0.016 (0.021) | 0.001 (0.003) |
| | P(Inhibit) | 0.5 (0.078) | 0.533 (0.019) |
| Non-rIFG | Mean SSD | 371 (81) | 399 (119) |
| | Go RT | 634 (61) | 644 (87) |
| | SSRT | 255 (47) | 238 (46) |
| | SR-RT | 552 (66) | 564 (93) |
| | P(Go miss) | 0.031 (0.033) | 0.04 (0.092) |
| | P(Go error) | 0.005 (0.007) | 0.004 (0.007) |
| | P(Inhibit) | 0.532 (0.013) | 0.536 (0.024) |

Mean SSD, Go RT, SSRT, and signal-respond RT (SR-RT) are presented on the ms scale. Go misses, Go errors, and Inhibition are given as probabilities ($0 \leq p \leq 1$). With the exception of P(Go error), all measures are computed after removing incorrect RTs and RTs faster than 200 ms, treating RTs slower than 1000 ms as censored observations. rIFG = inferior frontal gyrus; SSRT = stop-signal reaction time; SSD = stop-signal delay.

The same 2 × 3 ANOVA for the non-rIFG lesion group and their healthy matched comparisons only revealed the main effect of TRIAL TYPE ($F(2, 60) = 6.468$, p=0.003, $\eta^2{}_P$=0.177), with a comparable effect size to the rIFG group. However, unlike for the rIFG group, the interaction between GROUP and TRIAL TYPE and the main effect of GROUP were not significant (interaction: $F(2, 60) = 0.822$, p=0.445, $\eta^2{}_P$=0.027; GROUP: $F(1, 30) = 0.786$, p=0.382, $\eta^2{}_P = 0.026$). Furthermore, the same planned contrasts for the Successful-stop vs. Go difference revealed no difference in β-burst rate between non-rIFG lesion group ($M = 0.092$, SD = 0.092) and matched healthy comparison ($M = 0.064$, $SD = 0.125$) ($t(15) = 1.02$, p=0.324, $d = 0.255$). Finally, there was no significant difference in Failed-stop vs. Go difference between non-rIFG lesion group ($M = 0.074$, SD = 0.122) and their healthy comparisons ($M = 0.018$, SD = 0.157) ($t(15) = 1.497$, p=0.155, $d = 0.374$). Together, these results show that rIFG lesion patients show a significant reduction of stop-related frontal β-bursts compared to healthy comparisons, which was not the case for non-rIFG lesion patients.

We then also directly compared the Stop vs. Go differences in frontal β between the two lesion groups (*Appendix 1—figure 14A*). This comparison again showed a significantly reduced β-burst rate difference between Successful-stop and Go trials in the rIFG lesion group ($M = –0.005$, SD = 0.116) compared to the non-rIFG lesion group ($M = 0.092$, SD = 0.092) ($t(30) = –2.608$, p=0.014, $d = –0.922$, independent-samples $t$-test). The Failed-stop vs. Go difference showed the same pattern, with the rIFG lesion group ($M = –0.018$, SD = 0.106) showing a significantly reduced β-burst rate compared to the non-rIFG lesion group ($M = 0.074$, SD = 0.122) ($t(30) = –2.265$, p=0.031, $d = –0.801$, independent-samples $t$-test). All other group comparisons of frontal β-burst rate differences were added in Appendix 1 (*Appendix 1—figure 14*).

To sum up, these results confirmed our hypothesis that rIFG lesion patients show reduced β-burst rates on Successful-stop-trials compared to healthy comparisons, given their increase in stop-signal trigger failures. Moreover, this impairment was specific to the rIFG lesion group as no group difference was found between the non-rIFG lesion group and matched comparisons. Importantly, the rIFG group

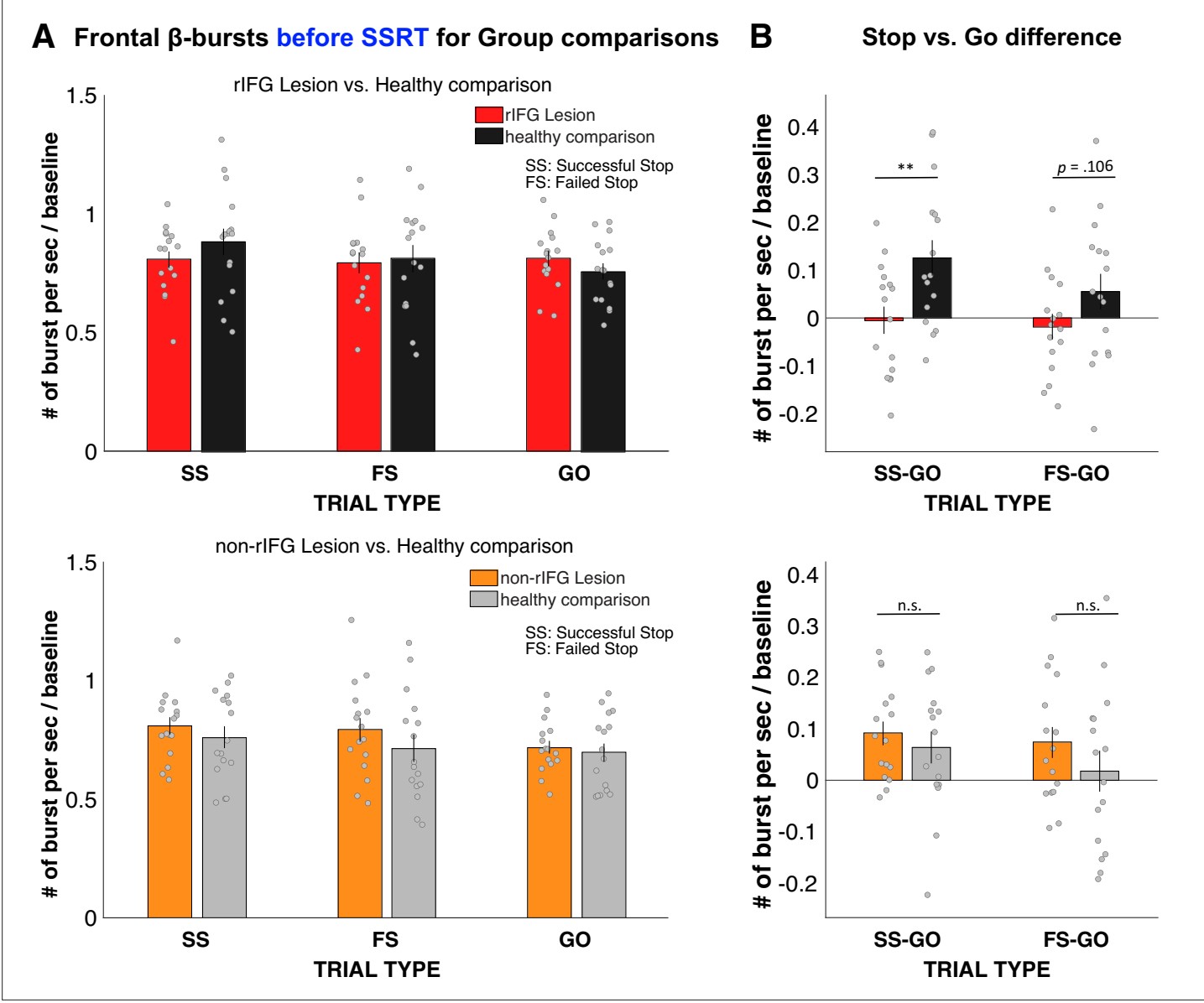

**Figure 3.** Normalized β-burst rate (per second) for group comparisons. (**A**) Frontal β-burst rate before stop-signal reaction time (SSRT) was shown for TRIAL TYPE (Successful-stop, Failed-stop, and Go) for two groups, respectively. Top: right inferior frontal gyrus (rIFG) lesion group vs. matched healthy comparison group. Bottom: non-rIFG lesion group vs. matched healthy comparison group. Dots represent individual participant means. Error bar indicates ± SEM. (**B**) Stop vs. Go difference (Successful-stop – Go and Failed-stop – Go). Top: rIFG lesion group vs. matched healthy comparison group. Bottom: non-rIFG lesion group vs. matched healthy comparison group. Dots represent individual participant means. Error bar indicates ± SEM.

also showed a significant reduction in frontal burst rates when directly compared to the non-rIFG lesion group.

## EEG results: Sensorimotor β-bursts after stop-related frontal bursts

We then investigated the temporal development of sensorimotor β-bursts after frontal β-bursts that occurred during the critical post-stop-signal window on Successful-stop trials (*Figure 4*). First, we replicated that across the whole sample, sensorimotor β-bursts were increased following frontal β-bursts (compared to trials without frontal β-bursts). This was done by comparing the sensorimotor β-burst rates on Successful-stop trials that contained frontal β-bursts in the stop-signal-to-SSRT period to that of trials without frontal β-bursts (in which sensorimotor β-bursts were time-locked to a random time point in the stop-signal-to-SSRT period) with paired-samples *t*-test at each time point between

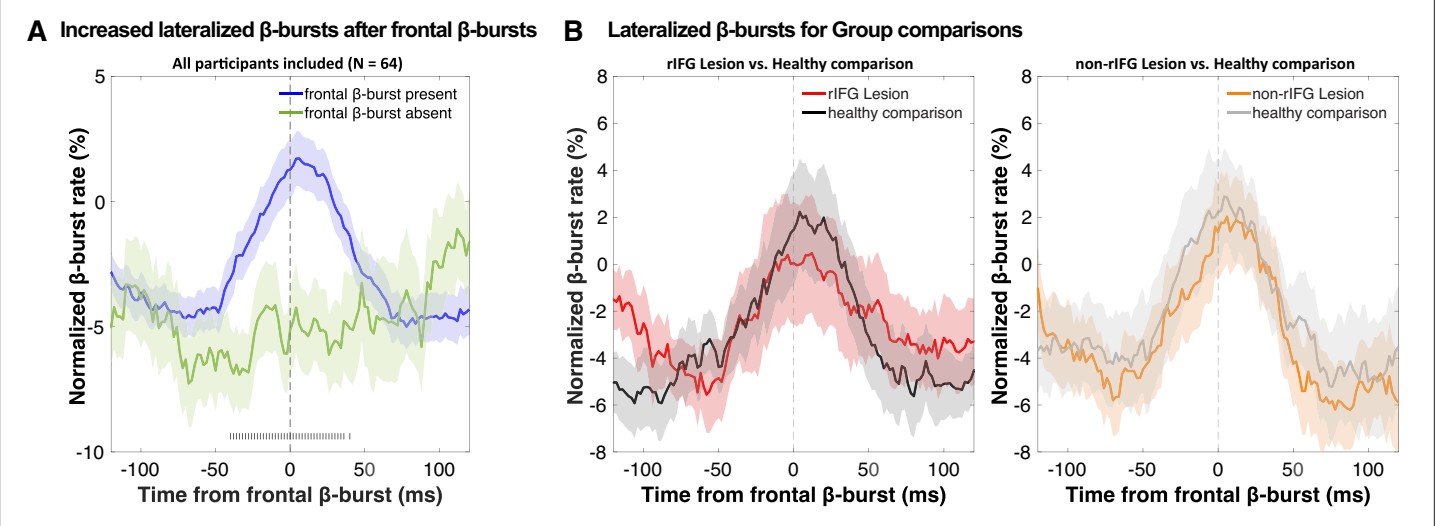

**Figure 4.** Normalized β-burst rate (%) at bilateral sensorimotor electrodes (C3/C4) following the first frontal β-burst in individual Successful-stop trials during the critical time period (stop-signal to stop-signal reaction time [SSRT]). (**A**) A comparison between trials with frontal β-bursts to trials without frontal β-bursts (where the sensorimotor β-bursts were time-locked to a random time point within a critical time period). The sensorimotor burst rates following frontal β-bursts were significantly increased (significance outlined at the bottom of the graph). The colored patch shows the ± SEM at each time point. (**B**) Group differences (lesion vs. matched healthy comparisons) in sensorimotor β-burst rates. No significant difference was found. The colored patch shows the ± SEM at each time point. Average trial count for these graphs varied between 57.7 and 63.5 and did not differ significantly between groups.

–100 and + 100 around the frontal β-bursts. Any significant difference between trials with frontal β-bursts and without frontal β-bursts is marked as gray hashes on the bottom of the figure (p<0.05). This analysis showed that sensorimotor β-burst rates were significantly increased following frontal β-bursts (*Figure 4A*), replicating *Wessel, 2020*. To test for potential group differences in this pattern, we then compared the sensorimotor burst rate between the respective lesion groups and their healthy comparisons (*Figure 4B*). No time point showed any significant differences at p<0.05, even absent any corrections for multiple comparisons. Together, this shows that the pattern of increased sensorimotor β-bursts following stop-related frontal β-burst was intact, even in the rIFG group.

## Discussion

In this study, we re-examined the role that rIFG plays in action-stopping and inhibitory control. rIFG has been at the center of what is arguably the most influential current neural theory of inhibitory control. The proposal that 'response inhibition can be localized to a discrete region within of the PFC' (*Aron et al., 2003*) has been highly influential (*Aron et al., 2004*). In its most recent iteration, the theory states that rIFG 'implements a brake over response tendencies' (*Aron et al., 2014*). Our work shows that this theory needs further revision. Indeed, rIFG does *not* seem to be primarily responsible for the *implementation* of inhibitory control. Instead, its role appears to be the triggering or *initiation* of the cascade that ultimately leads to the stopping of action. Concomitantly, in this study, rIFG patients showed more than fivefold increase in stop-signal trigger failures compared to matched healthy comparisons, as well as a significant reduction in stop-related β-bursts over frontal cortex (in fact, there did not seem to be *any* increase in frontal β-bursts in the rIFG group on stop compared to go-trials, strongly suggesting a causal role for rIFG in their generation). However, the implementation of inhibitory control itself appears to take place in other regions – and appears to remain intact in the rIFG group. In this study, this was suggested by the fact that in all groups (crucially, including rIFG patients), sensorimotor β-bursts were upregulated when a β-burst over frontal cortex did occur (i.e., when a purported initiation signal from frontal cortex was successfully sent).

There are at least two possible explanations for the finding that lesions to the rIFG cause such a substantial increase in stop-signal trigger failures. Both explanations imply substantially different roles for rIFG in cognitive control, and cognition more broadly. The first possible explanation is that the

function of rIFG is to specifically trigger inhibitory control during action-stopping or braking (e.g., in the types of situations that are simulated in the stop-signal task). In this framework, rIFG would still have an inhibition-specific role – though not in the actual implementation of the underlying process, but merely in its initiation. A different, competing explanation is that the function of rIFG is to detect *any* salient signal, and that it fulfills no role that is specific to inhibitory control. In the stop-signal task, rIFG would therefore merely be activated because of the saliency of the stop-signal, but not specifically because of the associated inhibitory requirements. In other words, it is up for debate whether the rIFG fulfills a role that is specific to inhibitory control or stopping/braking, or one that is more domain-general.

Along the lines of the latter explanation, rIFG is prominently at the core of another influential theory, which proposes that rIFG is part of a ventral attention network that functions as a 'circuit breaker.' In this framework, rIFG's role is to orient attention toward salient events – which would include, but not be limited to – stop-signals (*Corbetta et al., 2008*; *Corbetta and Shulman, 2002*). In line with this theory, fMRI work has shown that similar regions of rIFG are activated not just after stop-signals, but after any sort of salient, infrequent event (*Erika-Florence et al., 2014*; *Sharp et al., 2010*). As such, it is possible that the role of rIFG in stop-signal performance is indeed not specific to inhibitory control situations, but instead, that it merely detects the saliency of the stop-signal. In that scenario, the increase in trigger failures in the rIFG lesion group would reflect a more general attentional deficit, which – in the specific case of the stop-signal task – would express itself in the impaired detection of the stop-signal. However, while this impaired detection of the stop-signal would increase SSRT estimates, the underlying cause of that increase would be unrelated to the implementation of inhibitory control.

Crucially, however, any attempt to disentangle the domain-general detection of a salient stimulus from the stopping-specific implementation of inhibitory control is complicated by another factor. Namely, all salient stimuli, even those presented outside of stop-signal contexts, lead to an automatic, physiological inhibition of the motor system (*Dutra et al., 2018*; *Tatz et al., 2021*; *Wessel, 2018*; *Wessel and Aron, 2017*). For example, salient stimuli lead to a non-selective inhibition of corticomotor excitability (*Iacullo et al., 2020*), activate basal ganglia regions involved in inhibitory pathways (*Wessel et al., 2016*), reduce isometrically exerted force (*Novembre et al., 2018*; *Novembre et al., 2019*), and increase motoric response times (e.g., *Parmentier, 2008*). Indeed, it appears as if inhibitory control is a ubiquitous part of an organism's orienting response to salient stimuli (*Sokolov, 1963*). As such, inhibition and attention may be inextricably linked. If that is indeed the case, both competing theories of rIFG may be partially correct: the role of rIFG in cognitive control may indeed be domain-general, but its domain-general role may include the *triggering of inhibitory control following any type of salient stimulus*, specifically as part of a stereotypic and ubiquitous orienting response (*Wessel and Aron, 2017*). This possibility is explicitly taken into account in recent two-stage models of action-stopping (*Schmidt and Berke, 2017*; *Diesburg et al., 2021*). Either way, while it still remains to be seen whether the rIFG does have a specific role in inhibitory control or not, it seems safe to conclude that its role is not in the implementation of that process.

This latter fact is evident in this study by the fact rIFG patients could appropriately implement inhibitory control on the subset of trials in which it was ostensibly triggered successfully. This is primarily supported by the EEG-derived β-burst patterns. β activity in the LFP has long been linked to movement regulation (e.g., *Kilavik et al., 2013*; *Swann et al., 2009*; *Swann et al., 2011*). Recent work has shown that β activity in the human (and non-human) local field potential occurs in clearly demarcated, transient bursts (*Feingold et al., 2015*; *Sherman et al., 2016*; *Shin et al., 2017*). In acute stopping situations, a short-latency increase in such β-bursts over frontal cortex takes place (cf., *Figure 3*, see also *Enz et al., 2021*; *Jana et al., 2020*; *Wessel, 2020*). These frontal bursts are then followed by a short-latency increase in β-burst rates over sensorimotor cortex (cf., *Figure 4*, see also *Diesburg et al., 2021*; *Wessel, 2020*). Sensorimotor β has been long proposed to reflect an inhibited state of the motor system at rest (*Kilavik et al., 2013*; *Soh et al., 2021*). A re-instantiation of this sensorimotor β-bursting after frontal β-bursts has therefore been proposed to reflect the final stage of the inhibitory cascade during rapid action-stopping – the return of the motor system to its inhibited default state (*Wessel, 2020*). In the current study, we found that the initial frontal β-bursts immediately after the stop-signal were substantially reduced in the rIFG lesion group – both compared to healthy comparisons and compared to non-rIFG lesion patients. Conversely, however, all groups – notably

including rIFG lesion patients – showed a significant increase in sensorimotor β-bursting immediately after frontal β-bursts. In other words, in cases in which a frontal β-burst did take place in rIFG patients, their motor system was successfully returned to its inhibited state.

Together, this suggests that inhibitory control, while perhaps triggered in rIFG, is implemented in other areas. While there has been vigorous debate about the role of the rIFG (and other cortical regions, such as the pre-Supplementary Motor Area (SMA); *Nachev et al., 2007*) in action-stopping and inhibitory control, there seems to be some degree of consensus that the final steps on the way to successful action-stopping involve an interruption of thalamocortical motor representations via the output nuclei of the basal ganglia (see *Jahanshahi et al., 2015* for a review). This interruption is likely caused by the subthalamic nucleus (which is part of two long-hypothesized anti-kinetic cortico-basal ganglia-thalamocortical pathways: indirect and hyper-direct, cf., *Nambu et al., 2002*; *Parent and Hazrati, 1995*). In line with this, recent work has tracked the abovementioned β-burst dynamics further along the basal ganglia regions that are purportedly involved in the implementation of inhibitory control. Indeed, similar to the frontal β-burst rates report here and elsewhere, β-burst rates are also increased after stop-signals in both the subthalamic nucleus and the motor regions of the thalamus (*Diesburg et al., 2021*). Moreover, subthalamic bursts in particular are also followed by short-latency upregulations of β-bursts in sensorimotor areas, just like the frontal bursts in this study (and others). As such, it seems likely that while the rIFG is not directly involved in implementing inhibitory control, that function is fulfilled by the basal ganglia regions of the hyper and/or indirect pathways (*Schmidt et al., 2013*). To this point, recent work using electrical stimulation has shown that there is a direct, monosynaptic connection between rIFG and the subthalamic nucleus, and that this connection is highly relevant for the speed of inhibitory control (*Chen et al., 2020*). Indeed, even in healthy humans, variance in the integrity of this fiber tract directly map onto SSRT differences (*Coxon et al., 2012*). Together, a coherent picture emerges according to which rIFG, once activated by a salient signal (such as a stop-signal), triggers a multi-step cascade that culminates in successful action-stopping. At a minimum, this cascade involves the subthalamic nucleus, the output nuclei of the basal ganglia, and the motor thalamus, with the implementation of inhibitory control taking place in this subcortical chain. As these subcortical areas were intact in our rIFG lesion patients, it makes sense for the implementation of inhibitory control – once triggered – to be unimpaired in this population.

Due to its nature as a lesion study, the current work has some obvious limitations. First, while we believe that our lesions very closely resemble those in the original *Aron et al., 2003* study, no two samples of lesion patients are exactly the same. While the lesion in this study was somewhat more rostral compared to the Aron et al. work, both centers of gravity fell within the pars triangularis of rIFG. As such, we believe that our sample is as comparable as is possible within the constraints of human brain lesion studies. Second, as in every lesion study, damage was not limited to the region of interest. Regarding studies of rIFG in particular, other neighboring regions are typically affected by lesions as well – in particular, the anterior insula. That same region is notably active during action-stopping as well (e.g., *Boehler et al., 2010*), though recent work has shown that it has a more prominent role in failed rather than Successful-stop-trials (*Cai et al., 2014*). This is perhaps sensible, given the anterior insula's known role in action error processing (*Ullsperger et al., 2010*). On Successful-stop-trials, intracranial recordings have shown that anterior insula activity trails that of rIFG and likely occurs after SSRT (*Bartoli et al., 2018*). Together, these factors make it unlikely that coincidental damage to the anterior insula can account for the current set of findings.

Taken together, our data show that the dominant model of inhibitory control in the human brain needs to be fundamentally revised. Rather than the rIFG 'implementing a brake over response tendencies,' it appears as though its primary role is to detect salient signals (such as stop-signals), leading to the triggering of an inhibitory control process, with the implementation of the latter taking place in other areas.

## Materials and methods

### Participants

Sixty-four participants across four groups of N = 16 were recruited for the study, matching the sample size of the rIFG lesion group in the original *Aron et al., 2003* investigation. In addition to patients with focal lesions within the rIFG, we also included 16 non-rIFG lesion comparison patients – i.e.,

individuals with lesions outside of rIFG (see *Figure 1* for lesion overlap maps). All rIFG lesion and non-rIFG lesion patients were recruited from the Neurological Patient Registry of the University of Iowa's Division of Behavioral Neurology and Cognitive Neuroscience. Lesion etiologies included ischemic stroke (n = 13), hemorrhagic stroke (n = 4), focal contusion (n = 2), Arteriovenous Malformation (AVM) or cavernoma resection (n = 5), benign tumor resection (n = 3), herpes simplex encephalitis (n = 2), cyst resection (n = 1), abscess resection (n = 1), and epilepsy resection (n = 1). Patients taking psychoactive medications at dosages likely to induce cognitive side effects were not included. The rIFG and non-rIFG lesion groups varied somewhat regarding the chronicity of the lesion (*Table 1*). Three patients in the rIFG group had developmental-onset lesions. Exclusions of these patients did not affect the pattern of significance of the analyses of interest. Thirty-two age- and sex-matched healthy comparison participants were then also recruited from the Cognitive Neuroscience Registry for Normative Data of the Division of Behavioral Neurology and Cognitive Neuroscience and through local ads. All participants received detailed information describing the experiment and provided informed consent prior to participating in the study. The study was approved by the Institutional Review Board at the University of Iowa (IRB#201511709) and conducted in accordance with the Declaration of Helsinki. Data collection was performed in an EEG laboratory in the Neurology Clinic at the University of Iowa between October 2018 and July 2021. All lesion patients were compensated at an hourly rate of $30, while healthy participants were compensated at an hourly rate of $15. Monetary compensations for mileage and meal were also provided. Demographic data for all participants are presented in *Table 1*.

## Task and procedure

Participants performed a stop-signal task presented using Psychtoolbox (version 3, *Brainard, 1997*) in MATLAB 2015b (The MathWorks, Natick, MA) on an Ubuntu Linux desktop computer. Responses were made using a standard QWERTY USB keyboard. A schematic design of the stop-signal task is shown in *Figure 2*. Each trial began with a fixation cross at the center of the screen (500 ms) followed by a black arrow (Go-signal) pointing either left or right, displayed for 1000 ms. Two white stickers were attached on the 'q' key and 'p' key in the keyboard to indicate Go stimulus-response key mapping. Participants were instructed to press the left key (the 'q' key) with their left index finger in response to the left arrow and the right key (the 'p' key) with their right index finger in response to the right arrow as fast and accurately as possible. In five rIFG lesion patients (as well as their respective matched comparison participants), participants instead made unimanual responses with two fingers of their right hand using the arrow keys on the keyboard. This was due to lesions to the right hemisphere of the lesion patients that encompassed motor cortex and surrounding areas, which could affect the mobility of their left hand.

If no response was made during the Go-signal presentation (1000 ms), the feedback 'Too Slow!' was presented for 1000 ms at the center of the screen. Note that no response times were collected after the end of this 1000 ms window for rIFG patients and their healthy comparisons, as well as for four non-rIFG lesion patients and two healthy comparisons (the procedure was then slightly changed to keep recording responses even after this window, though those responses were still counted as misses for the purposes of the behavioral analyses). While our ex-Gaussian approach (see below) was explicitly designed to handle the potential censoring of the Go-RT distributions resulting from this cutoff deadline, we still recollected two rIFG lesion patients' data, whose Go-RT distributions showed signs of censoring.

On one-third of trials, an auditory Stop-signal (900 Hz sine-wave tones of 100 ms duration) was presented after the Go-signal at a varying stop-signal delay (SSD). Participants were instructed to withhold their response on such trials. The SSD was initially set to 200 ms and adjusted separately for left and right responses depending on stop success (50 ms increment) or failure (50 ms decrement), with a goal of achieving an overall p(stop) of approximately 0.50 (*Verbruggen et al., 2019*). The overall trial length was fixed to 3500 ms. Before the experiment, participants performed a short practice block (24 trials). In total, participants underwent 480 trials (eight blocks of 60 trials; 320 go/160 stop-trials overall). In each rest period between blocks, performance feedback was given on the previous block.

## Bayesian modeling of behavioral data

The stop-signal data were analyzed with BEESTS, a hierarchical Bayesian modeling technique that simultaneously accounts for the shape of Go-RT and SSRT distributions and the prevalence of trigger

failures in the stop-signal task (*Matzke et al., 2013*; *Matzke et al., 2017b*). As shown in *Figure 2B*, BEESTS is based on the horse-race model (*Logan and Cowan, 1984*) and assumes that response inhibition depends on the relative finishing times of a go and a stop runner, triggered by the go and the stop stimuli, respectively. On a given trial, if the go runner is slower than SSD + the finishing time of the stop runner, the go response is successfully stopped (i.e., the stop process wins the race). If the go runner is faster than SSD + the finishing time of the stop runner, inhibition fails and a signal-respond RT (e.g., gray-color distribution in *Figure 2*) is produced. BEESTS assumes that the finishing times of the go (Go-RT distribution) and the stop runners (SSRT distribution) follow an ex-Gaussian distribution with parameters $\mu$, $\sigma$, and $\tau$. The $\mu$ and $\sigma$ parameters reflect the mean and the standard deviation of the Gaussian component, and $\tau$ gives the mean of the exponential component and reflects the slow tail of the distribution. The mean and variance of the finishing time distributions can be obtained as $\mu + \tau$ (i.e., mean Go-RT and SSRT) and $\sigma^2 + \tau^2$, respectively. Using a mixture-likelihood approach, the model can be augmented with a parameter, P(*TF*), that quantifies the probability that participants fail to trigger the stop runner (*Matzke et al., 2017b*).

Incorrect RTs and RTs faster than 200 ms (e.g., anticipatory responses) were removed before fitting the data. In total, this excluded 111 trials across the rIFG group (1.45% of all data), 15 trials in the non-rIFG group (0.2% of all data), and 88 trials in the matched comparisons (0.55% of all data). We treated omissions on go trials and RTs on go and stop-signal trials slower than 1000 ms as censored observations and adjusted the likelihood of the model to account for the upper censoring of the finishing time distributions. We modeled the data of each group separately using a single go runner and a stop runner, resulting in seven parameters per participant: $\mu_{go}$, $\sigma_{go}$, and $\tau_{go}$ for the go runner, and $\mu_{stop}$, $\sigma_{stop}$, $\tau_{stop}$, and P(TF) for the stop runner. The P(TF) parameter was estimated on the real line after transformation from the probability scale using a probit transformation. We did not use the recently extended version of the model (which uses separate go runners corresponding to the two go response options) as direction error responses were very rare (*Matzke et al., 2019*). Note also that the 'censored' BEESTS model, in its current form, does not allow for the estimation of 'go failures,' i.e., the probability that participants fail to trigger the go runner.

We assumed (truncated) normal population-level distributions for all model parameters, including the probit transformed P(TF) parameter. The population-level distributions were parameterized and estimated in terms of their location and scale parameters, which were then transformed back to means and standard deviations for inference. As shown in Appendix 1, we assigned weakly informative priors to the population-level location and scale parameters that covered a wide but realistic range (e.g., *Matzke et al., 2019*).

The analyses were carried out in the Dynamic Models of Choice software (*Heathcote et al., 2019*) using Differential Evolution Markov chain Monte Carlo (DE-MCMC; *TerBraak, 2006*) sampling implemented in the R programming environment (*R Core Team, 2015*). We used parameter estimates obtained from fitting each participant's data individually using non-hierarchical Bayesian estimation as start values for the hierarchical sampling routine. The number of MCMC chains was set to 21, i.e., three times the number of participant-level model parameters. To reduce autocorrelation, the MCMC chains were thinned to retain only every 15th draw from the joint posterior distribution. During the burn-in period, the probability of a migration step was set to 5%, after which only crossover steps were performed. Convergence was assessed using univariate and multivariate proportional scale-reduction factors ($R < 1.1$; *Brooks and Gelman, 1998*; *Gelman and Rubin, 1992*) and visual inspection of the MCMC chains.

The absolute goodness-of-fit of the model was assessed with posterior predictive model checks (*Gelman et al., 1996*) using the average cumulative distribution function of Go-RTs and signal-respond RTs, inhibition functions, and median signal-respond RTs as a function of SSD. Decision about the descriptive accuracy of the model was based on visual inspection of the model predictions, aided with posterior predictive p-values (for details, see *Matzke et al., 2019*). As shown in Appendix 1, the model provided a good account of all these aspects of the observed data for the rIFG lesion patients and the matched healthy comparison participants (*Appendix 1—figure 1–4*), but it showed a quantitatively small misfit (i.e., underprediction) to the average inhibition function at short SSDs in the other two groups (*Appendix 1—figures 6 and 8*). To examine the robustness of the results to a possible model misspecification, we sequentially removed all stop-signal trials from the data of the non-rIFG lesion and the matched comparison group at SSDs of 0, 50, 100, 150, and 250 ms (i.e., SSDs

where misfit occurred), refit the model, and reassessed the model's descriptive accuracy. Descriptive accuracy improved as stop-signal trials on short SSDs were removed (*Appendix 1—figures 10 and 12*). Importantly, qualitative conclusions about group differences were the same whether or not stop trials with short SSDs were included in the analysis. This indicates the robustness of the results and supports the validity of our conclusions. To avoid overconfidence and ensure that our conclusions are based on a descriptively accurate model, we report results based on mixing the posterior distributions estimated using the full data set and the five subsets after removing SSDs between 0 and 250 ms.

## Statistical inference on model parameters

We used the mean of the posterior distributions as point estimates for the BEESTS model parameters, and the 2.5th and 97.5th percentile of the distributions (i.e., 95% credible interval [CI]) to quantify estimation uncertainty. Inference about group differences was based on the overlap between the posterior distributions of the population-level parameters of the different groups (i.e., both lesion groups vs. their respective matched healthy comparison group). Overlap was quantified using Bayesian p values computed as the proportion of samples in the posterior distribution of the lesion group that is larger than in the matched comparison group. Bayesian p values close to 0 or 1 indicate that the posterior distribution of the lesion group is shifted to lower or higher values, respectively, relative to the matched comparison group, suggesting the presence of a group difference. Bayesian p values were computed after appropriate transformations of the posterior distributions (i.e., bivariate inverse probit transformation for the P(TF) parameter and transformation of the truncated normal population-level location parameters to means). The posterior distributions of the population-level mean Go-RT and SSRT were obtained by computing $\mu_{go} + \tau_{go}$ and $\mu_{stop} + \tau_{stop}$, respectively, for each MCMC iteration and then collapsing the resulting population-level samples in a single distribution across chains. Similarly, the posterior distributions of the participant-level mean Go-RT and SSRT were computed by summing the corresponding participant-level $\mu$ and $\tau$ samples for each iteration and collapsing the resulting samples across the chains. The point estimates used in the analysis of β-burst events reflect the mean of the participant-level posteriors.

The preregistration document for these analyses can be found at https://osf.io/d9r4s/.

## EEG recording

Sixty-four-channel EEG data in the extended 10–10 system were recorded using two Brain Products systems (actiChamp or MRplus). Ground and reference electrodes were placed at AFz and Pz, respectively. The MRplus system included two additional electrodes on the left canthus (for the horizontal eye movement) and below the left eye (for the vertical eye movement). EEG data was digitized with a sampling rate of 500 Hz, with hardware filters set to 10 s time-constant high-pass and 1000 Hz low-pass.

## EEG preprocessing

EEG data preprocessing was conducted using custom routines in MATLAB. The data were filtered (high-pass cutoff: 0.3 Hz; low-pass cutoff: 50 Hz) and then visually inspected to identify and remove non-stereotypical artifacts. The data were subsequently re-referenced to common average and subjected to a temporal infomax independent component analysis (ICA; *Bell and Sejnowski, 1995*) as implemented in EEGLAB (*Delorme and Makeig, 2004*). Components representing stereotypic artifact activity (saccades, blinks, and electrode artifacts) were identified using outlier-based statistics and were removed from the data. We further obtained dipole solutions for ICs using the Dipfit plug-in for EEGLAB and further rejected ICs with residual variance larger than 15%, which typically represent non-brain data (*Delorme et al., 2012*). The remaining components were backprojected into channel space to reconstruct artifact-free channel data and subjected to further analyses. Finally, the channel-space data were then transformed using the current-source density method (CSD; *Kayser and Tenke, 2006*), which attenuates the effects of volume conduction on the scalp-measured activity.

## β-Burst detection

β-Burst detection was performed using the same method as described in *Wessel, 2020*, except with a burst detection threshold of 2× median power (rather than 6× median), following the recent

recommendation from **Enz et al., 2021**. First, each electrode's data were convolved with a complex Morlet wavelet of the form:

$$w\left(t,f\right) = A \exp\left(-\frac{t^2}{2\sigma_t^2}\right) \exp\left(2i\pi ft\right)$$

With $\sigma = \frac{m}{2\pi f}$, $A = \frac{1}{\sigma_t}\sqrt{2\pi}$, and m = 7 (cycles) for each of the 15 evenly spaced frequencies spanning the β-band (15–29Hz). Time-frequency power estimates were extracted by calculating the squared magnitude of the complex wavelet-convolved data. These power estimates were then epoched relative to the events in question (ranging from 500 – 1000 ms with respect to Stop-/Go-signals). Individual β-bursts were defined as local maxima (using the MATLAB function *imregionalmax*) in the trial-by-trial β-band time-frequency power matrix for which the power exceeded a cutoff of 2× the median power of the entire time-frequency power matrix for that electrode (**Enz et al., 2021**).

## Statistical analysis of β-burst events

The quantification of frontal β-bursts was done as in **Wessel, 2020**. β-Burst events in the critical time period ranging from the stop-signal onset to each individual's SSRT estimate were counted separately for successful- and Failed-stop trials at electrode FCz. For Go trials, we counted the number of β-burst events in a time period of identical length, ranging from the current SSD on the trial and the participants' SSRT estimate (in other words, in the time period during which a stop-signal would have appeared on that particular trial and the end of SSRT). We then converted these numbers to β-burst rate (bursts per second) by dividing each participant's burst rates with the length of individual SSRT estimate. We then normalized each participant's burst/s measurement with a baseline time period of [–500 0] relative to the Go stimulus onset on the trial for each trial type (Successful-stop, Failed-stop, and Go). This normalization procedure was adapted from **Jana et al., 2020**. However, the normalization had no significant influence on the results, and all significances remained intact when raw β-burst rates were investigated. These normalized β-burst rates were then analyzed with 2-by-3 mixed ANOVA with between-subjects factor of LESION (lesion vs. matched healthy comparison) and within-subjects factor of TRIAL TYPE (Successful-stop, Failed-stop, Go). This was done separately for the rIFG lesion group and the non-rIFG lesion group. Planned comparisons for Stop vs. Go difference (e.g., Successful-stop vs. Go and Failed-stop vs. Go) within the groups (e.g., lesion vs. matched healthy comparison) were made using paired-samples *t*-tests.

Furthermore, we investigated the implementation of inhibitory control at sensorimotor sites following frontal β-bursts. To this end, we first identified Successful-stop-trials in which at least one frontal β-burst event occurred within the critical time period (i.e., between SSD and SSD + SSRT). Next, we time-locked the data to the first frontal β-burst event within that period and quantified the sensorimotor β-burst rate at C3/C4 electrodes from –100 to 100 ms around that frontal burst. In past investigations, this showed clearly increases in β-burst rate immediately (within ~25 ms) following frontal bursts, which are not present during matched time periods on trials without frontal bursts (e.g., **Diesburg et al., 2021**; **Enz et al., 2021**; **Wessel, 2020**). In this study, instead of making arbitrary time-bins, we used a sliding-window approach, with a search window of 50 ms around each sample point (±25 ms). This method avoids the inherently arbitrary nature of the binning approach.

We first transformed the mean sensorimotor β-burst rates at each of these time points to percent change from baseline, with the baseline being the mean β-burst rate in the 500 ms pre-GO signal period of the same trial. Next, these normalized β-burst rates at each time point were compared between groups (lesion vs. matched healthy comparison) using sample-wise paired-samples *t*-tests covering the whole time period from –100 to 100 ms around the frontal burst.

## Acknowledgements

The authors acknowledge the National Institutes of Health grant R01 NS117753 to JRW and Netherlands Organization of Scientific Research (NWO) Vidi grant (VI.Vidi.191.091) to DM.

# Additional information

### Funding

| Funder | Grant reference number | Author |
|---|---|---|
| National Institutes of Health | R01 NS117753 | Jan R Wessel |
| Nederlandse Organisatie voor Wetenschappelijk Onderzoek | VI.Vidi.191.091 | Dora Matzke |

The funders had no role in study design, data collection and interpretation, or the decision to submit the work for publication.

### Author contributions

Yoojeong Choo, Formal analysis, Investigation, Visualization, Writing - review and editing; Dora Matzke, Conceptualization, Software, Formal analysis, Investigation, Methodology, Writing - review and editing; Mark D Bowren Jr, Formal analysis, Investigation, Visualization; Daniel Tranel, Data curation, Validation, Methodology, Writing - review and editing; Jan R Wessel, Conceptualization, Supervision, Funding acquisition, Investigation, Methodology, Writing - original draft, Project administration, Writing - review and editing

### Author ORCIDs

Yoojeong Choo http://orcid.org/0000-0003-1261-9129
Jan R Wessel http://orcid.org/0000-0002-7298-6601

### Ethics

Human subjects: All participants received detailed information describing the experiment and provided informed consent prior to participating in the study. The study was approved by the Institutional Review Board at the University of Iowa (IRB#201511709) and conducted in accordance with the Declaration of Helsinki.

### Decision letter and Author response

Decision letter https://doi.org/10.7554/eLife.79667.sa1
Author response https://doi.org/10.7554/eLife.79667.sa2

# Additional files

### Supplementary files

• MDAR checklist

### Data availability

All data and analysis scripts can be downloaded at the following URL: https://osf.io/ck5zd/.

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

## Appendix 1

### Prior distributions

We modeled the parameters of each participant $j$, $j$ = 1,..., 16, in each group using (truncated) normal population-level distributions, with location (M), scale (S), and lower and upper bounds as specified below:

$$\mu_{go,j} \sim \text{Truncated Normal}(M\mu_{go}, S\mu_{go})[0, 4],$$
$$\sigma_{go,j} \sim \text{Truncated Normal}(M\sigma_{go}, S\sigma_{go})[0, 4],$$
$$\tau_{go,j} \sim \text{Truncated Normal}(M\tau_{go}, S\tau_{go})[0, 4],$$
$$\mu_{stop,j} \sim \text{Truncated Normal}(M\mu_{stop}, S\mu_{stop})[0, 4],$$
$$\sigma_{stop,j} \sim \text{Truncated Normal}(M\sigma_{stop}, S\sigma_{stop})[0, 4],$$
$$\tau_{stop,j} \sim \text{Truncated Normal}(M\tau_{stop}, S\tau_{stop})[0, 4],$$
$$TF_j \sim \text{Normal}(M_{TF}, S_{TF}).$$

Note that the participant-level P(TF) parameter was first projected from the probability scale to the real line with a probit transformation.

We used (truncated) normal prior distributions for the population-level location parameters:

$$M\mu_{go}, M\mu_{stop} \sim \text{Truncated Normal}(0.5, 1)[0, 4]$$
$$M\sigma_{go}, M\tau_{go}, M\sigma_{stop}, M\tau_{stop} \sim \text{Truncated Normal}(0.1, 1)[0, 4]$$
$$M_{TF} \sim \text{Normal}(-1.5, 1)$$

We used exponential prior distributions for the population-level scale parameters:

$$S\mu_{go}, S\sigma_{go}, S\tau_{go}, S\mu_{stop}, S\sigma_{stop}, S\tau_{stop}, S_{TF} \sim \text{Exponential}(1)$$

As the data were fit in seconds, the priors are also parameterized on the second scale.

### Goodness-of-fit

We assessed the descriptive accuracy of the 'censored' BEESTS model by comparing the observed data to predictions based on the joint posterior distribution of the participant-level parameters. For each group, we used 500 randomly selected parameter vectors from the participant-level joint posterior to generate 500 predicted stop-signal data sets per participant using the observed SSD and the observed number of go and stop-signal trials. We focused on three aspects of the data: the average cumulative distribution function of Go-RTs and signal-respond RTs, inhibition functions, and median signal-respond RTs as a function of SSD. The model provided a good account of all these aspects of the data for the rIFG lesion patients and the matched healthy comparison participants (*Appendix 1—figure 1–4*), but it showed a quantitatively small misfit (i.e., underprediction) to the average inhibition function at short SSDs in the other two groups (*Appendix 1—figures 6 and 8*). Descriptive accuracy in the non-rIFG lesion and matched comparison group improved after stop-signal trials on short SSDs were removed (*Appendix 1—figures 10 and 12*): after removing stop trials at SSDs with 0–250 ms, the model was able to accurately account for the observed inhibition function in both groups.

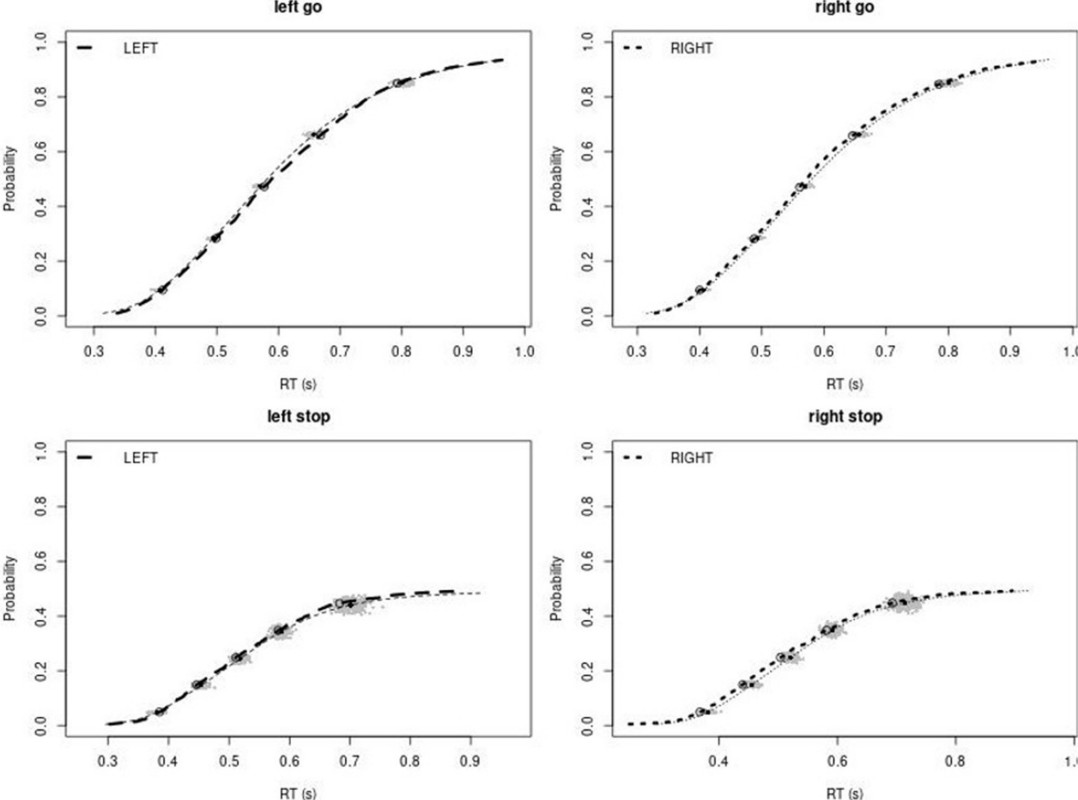

**Appendix 1—figure 1.** Observed and predicted cumulative distribution function (CDF) of Go-RTs (upper panels) and signal-respond RTs (lower panels), separately for left and right stimuli, for the right inferior frontal gyrus (rIFG) lesion group. The observed and predicted CDFs were averaged across participants. Signal-respond RTs were collapsed across stop-signal delay (SSD). Thick dashed and dotted lines show the CDF of the observed 'LEFT' and 'RIGHT' responses, respectively. Circles show the 10th, 30th, 50th, 70th, and 90th percentile of the distributions. Thin dashed and dotted lines show the CDF of the predicted 'LEFT' and 'RIGHT' responses, respectively, averaged across the 500 predictions. For each percentile, the gray clouds show the 500 predicted percentiles.

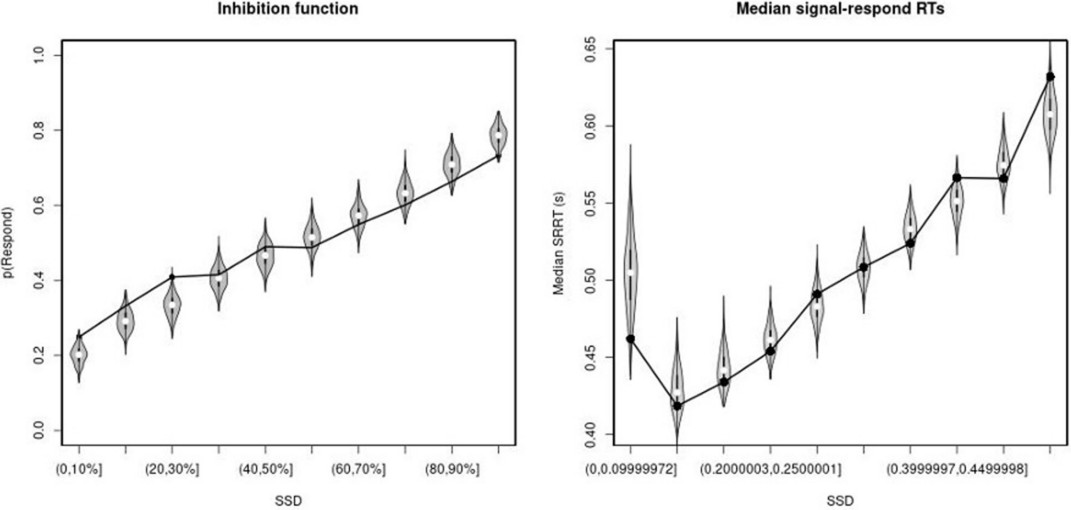

**Appendix 1—figure 2.** Observed and predicted inhibition function (left panel) and median signal-respond RT as a function of stop-signal delay (SSD; right panel) for the right inferior frontal gyrus (rIFG) lesion group. In the left panel, black bullets show the observed average response rate on stop-signal trials (P(Respond)) for each SSD category, where the SSD categories were defined in terms of the percentiles of the distribution of SSDs

*Appendix 1—figure 2 continued on next page*

*Appendix 1—figure 2 continued*

for each participant and then averaged across participants. In the right panel, black bullets show the observed average median signal-respond RT (SRRT) for each SSD category, where SSD categories were defined by pooling SSDs over participants before calculating the percentiles. The gray violin plots show the distribution of the 500 average response rates and SRRTs predicted by the model, with the white circles representing the median of the predictions.

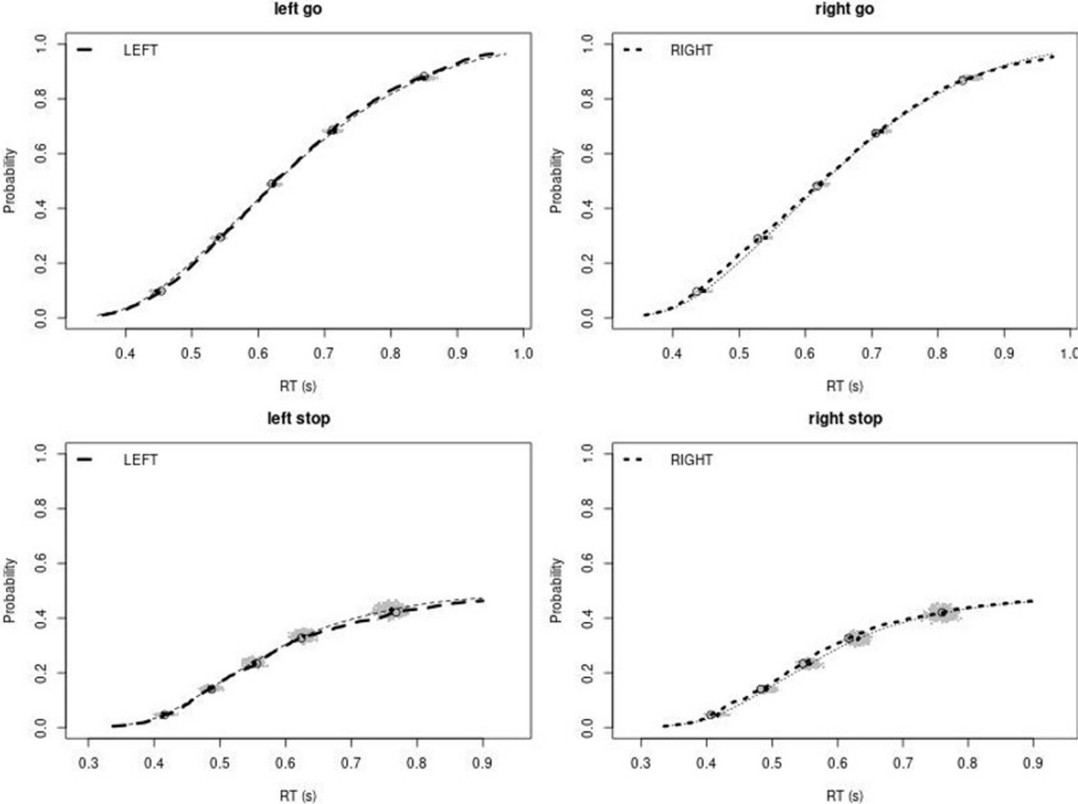

**Appendix 1—figure 3.** Observed and predicted cumulative distribution function (CDF) of Go-RTs (upper panels) and signal-respond RTs (lower panels), separately for left and right stimuli, for the matched comparison group for the right inferior frontal gyrus (rIFG) lesion patients. The observed and predicted CDFs were averaged across participants. Signal-respond RTs were collapsed across SSD. Thick dashed and dotted lines show the CDF of the observed 'LEFT' and 'RIGHT' responses, respectively. Circles show the 10th, 30th, 50th, 70th, and 90th percentile of the distributions. Thin dashed and dotted lines show the CDF of the predicted 'LEFT' and 'RIGHT' responses, respectively, averaged across the 500 predictions. For each percentile, the gray clouds show the 500 predicted percentiles.

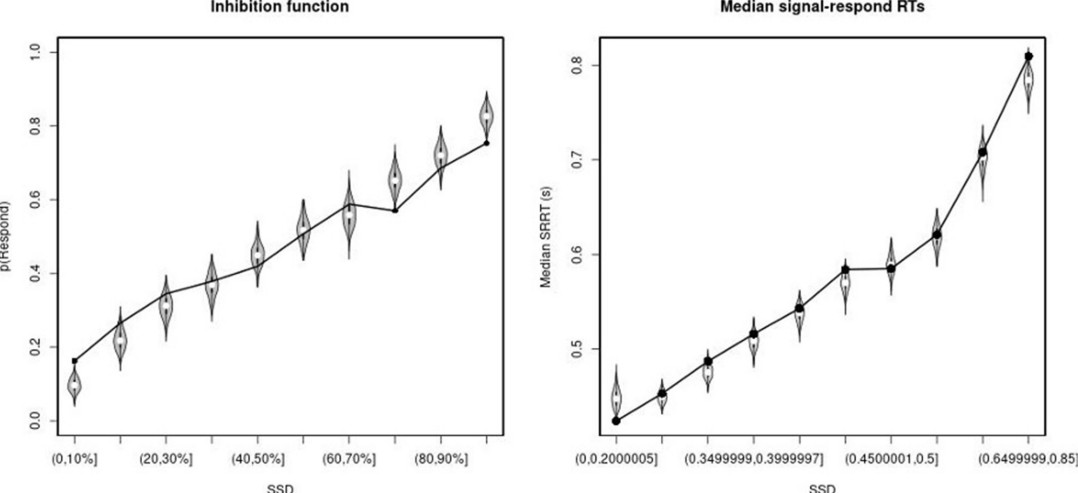

**Appendix 1—figure 4.** Observed and predicted inhibition function (left panel) and median signal-respond RT as a function of stop-signal delay (SSD; right panel) for the matched comparison group for the right inferior frontal gyrus (rIFG) lesion patients. In the left panel, black bullets show the observed average response rate on stop-signal trials (P(Respond)) for each SSD category, where the SSD categories were defined in terms of the percentiles of the distribution of SSDs for each participant and then averaged across participants. In the right panel, black bullets show the observed average median signal-respond RT (SRRT) for each SSD category, where SSD categories were defined by pooling SSDs over participants before calculating the percentiles. The gray violin plots show the distribution of the 500 average response rates and SRRTs predicted by the model, with the white circles representing the median of the predictions.

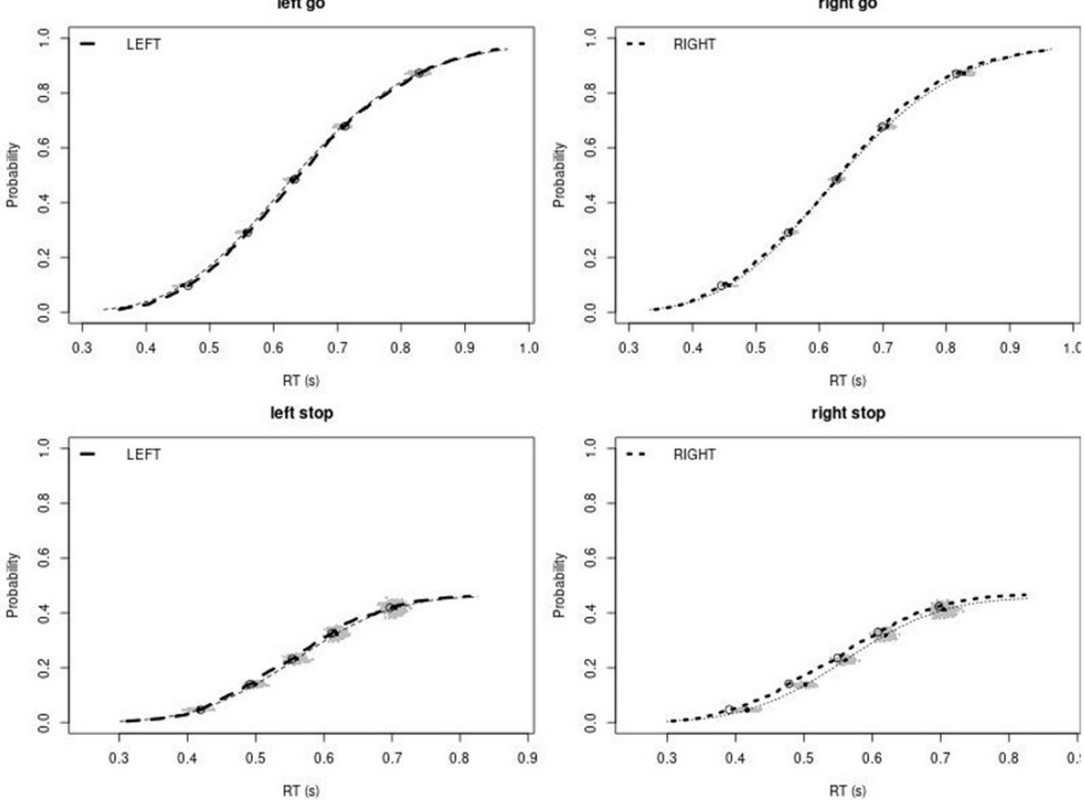

**Appendix 1—figure 5.** Observed and predicted cumulative distribution function (CDF) of Go-RTs (upper panels) and signal-respond RTs (lower panels), separately for left and right stimuli, for the non-right inferior frontal gyrus
*Appendix 1—figure 5 continued on next page*

(non-rIFG) lesion group. The observed and predicted CDFs were averaged across participants. Signal-respond RTs were collapsed across SSD. Thick dashed and dotted lines show the CDF of the observed 'LEFT' and 'RIGHT' responses, respectively. Circles show the 10th, 30th, 50th, 70th, and 90th percentile of the distributions. Thin dashed and dotted lines show the CDF of the predicted 'LEFT' and 'RIGHT' responses, respectively, averaged across the 500 predictions. For each percentile, the gray clouds show the 500 predicted percentiles.

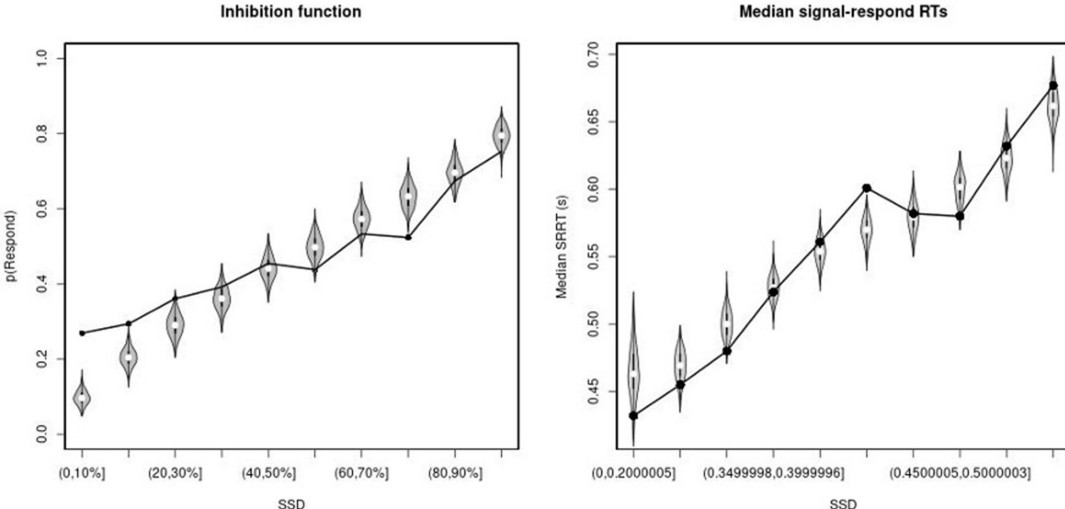

**Appendix 1—figure 6.** Observed and predicted inhibition function (left panel) and median signal-respond RT as a function of stop-signal delay (SSD; right panel) for the non-right inferior frontal gyrus (non-rIFG) lesion patients. In the left panel, black bullets show the observed average response rate on stop-signal trials (P(Respond)) for each SSD category, where the SSD categories were defined in terms of the percentiles of the distribution of SSDs for each participant and then averaged across participants. In the right panel, black bullets show the observed average median signal-respond RT (SRRT) for each SSD category, where SSD categories were defined by pooling SSDs over participants before calculating the percentiles. The gray violin plots show the distribution of the 500 average response rates and SRRTs predicted by the model, with the white circles representing the median of the predictions.

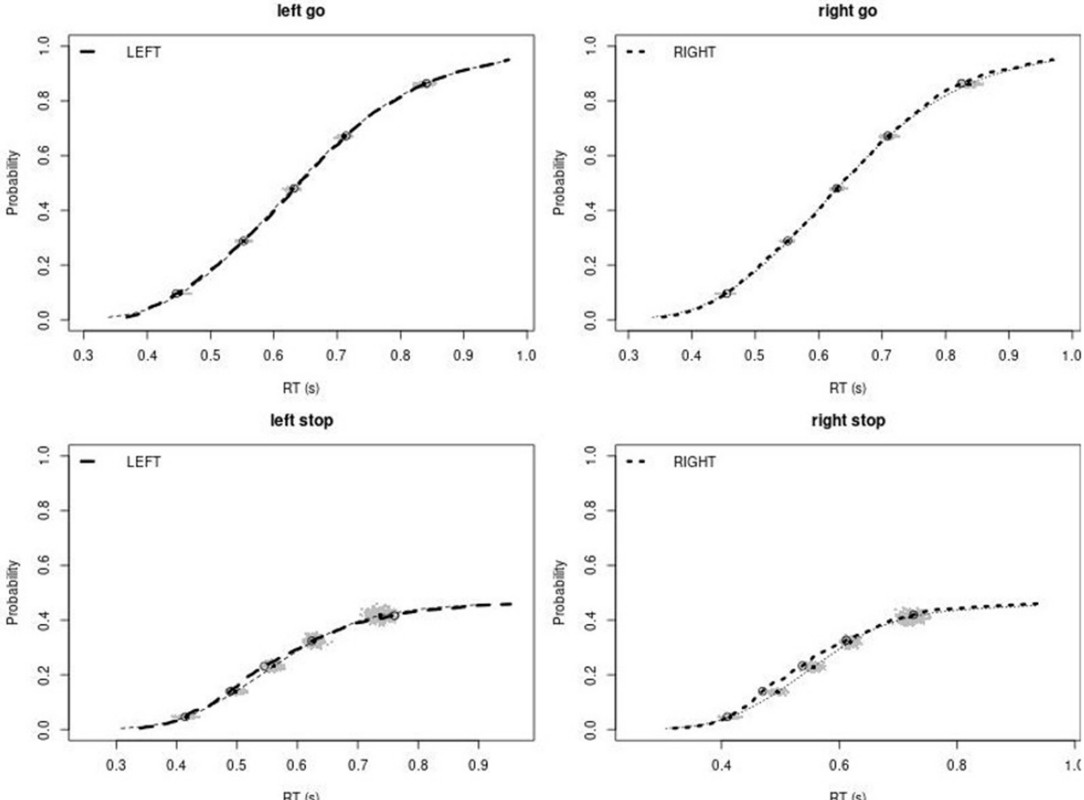

**Appendix 1—figure 7.** Observed and predicted cumulative distribution function (CDF) of Go-RTs (upper panels) and signal-respond RTs (lower panels), separately for left and right stimuli, for the matched comparison group for the non-right inferior frontal gyrus (non-rIFG) lesion patients. The observed and predicted CDFs were averaged across participants. Signal-respond RTs were collapsed across stop-signal delay (SSD). Thick dashed and dotted lines show the CDF of the observed 'LEFT' and 'RIGHT' responses, respectively. Circles show the 10th, 30th, 50th, 70th, and 90th percentile of the distributions. Thin dashed and dotted lines show the CDF of the predicted 'LEFT' and 'RIGHT' responses, respectively, averaged across the 500 predictions. For each percentile, the gray clouds show the 500 predicted percentiles.

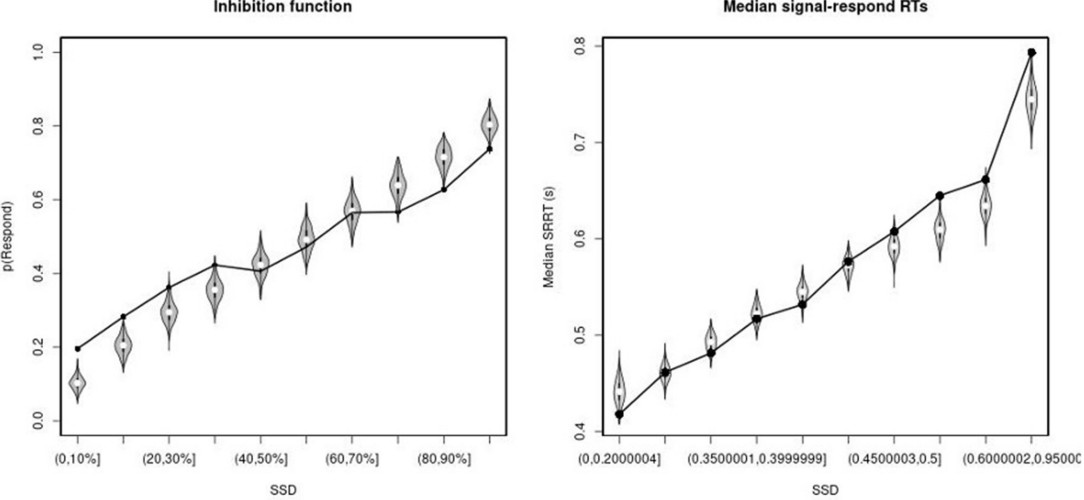

**Appendix 1—figure 8.** Observed and predicted inhibition function (left panel) and median signal-respond RT as a function of stop-signal delay (SSD; right panel) for the matched comparison group for the non-right inferior frontal gyrus (non-rIFG) lesion patients. In the left panel, black bullets show the observed average response rate
*Appendix 1—figure 8 continued on next page*

*Appendix 1—figure 8 continued*
on stop-signal trials (P(Respond)) for each SSD category, where the SSD categories were defined in terms of the percentiles of the distribution of SSDs for each participant and then averaged across participants. In the right panel, black bullets show the observed average median signal-respond RT (SRRT) for each SSD category, where SSD categories were defined by pooling SSDs over participants before calculating the percentiles. The gray violin plots show the distribution of the 500 average response rates and SRRTs predicted by the model, with the white circles representing the median of the predictions.

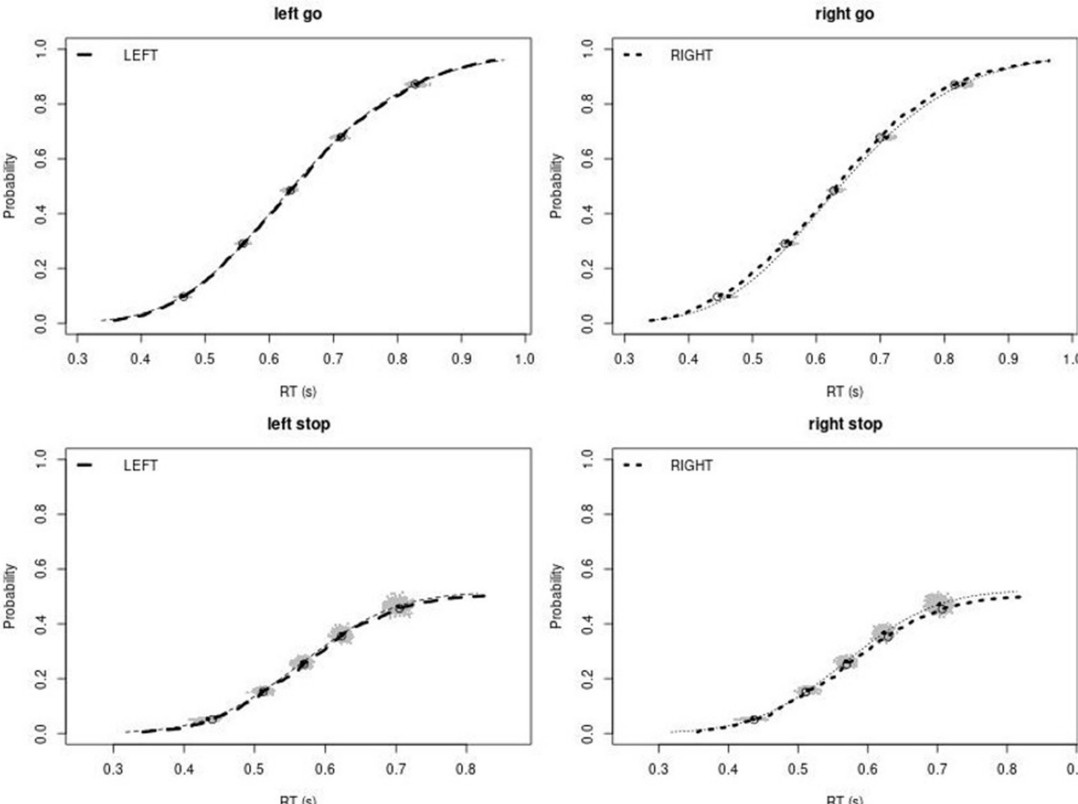

**Appendix 1—figure 9.** Observed and predicted cumulative distribution function (CDF) of Go-RTs (upper panels) and signal-respond RTs (lower panels), separately for left and right stimuli, for the non-right inferior frontal gyrus (non-rIFG) lesion group after removing stop-signal trials at stop-signal delays (SSDs) of 0, 50, 100, 150, and 250 ms. The observed and predicted CDFs were averaged across participants. Signal-respond RTs were collapsed across SSD. Thick dashed and dotted lines show the CDF of the observed 'LEFT' and 'RIGHT' responses, respectively. Circles show the 10th, 30th, 50th, 70th, and 90th percentile of the distributions. Thin dashed and dotted lines show the CDF of the predicted 'LEFT' and 'RIGHT' responses, respectively, averaged across the 500 predictions. For each percentile, the gray clouds show the 500 predicted percentiles.

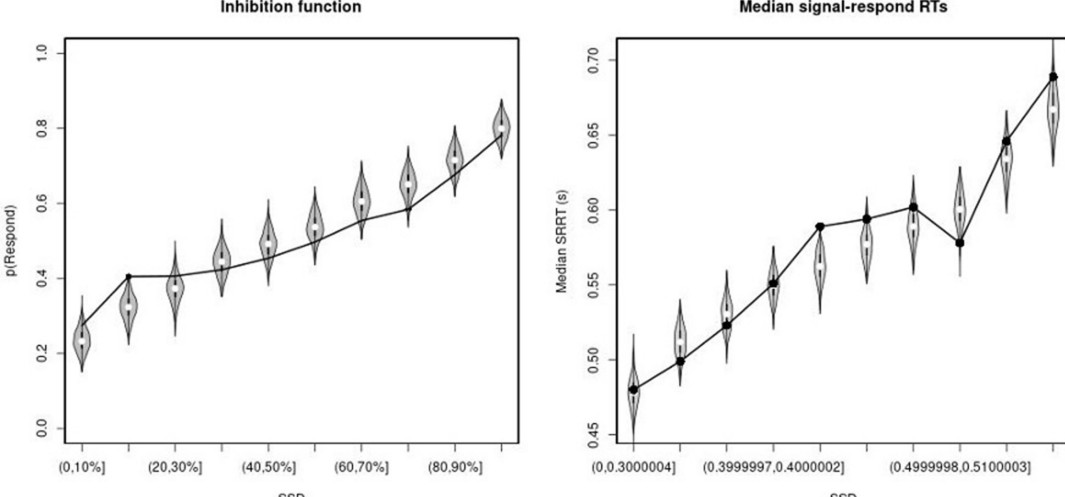

**Appendix 1—figure 10.** Observed and predicted inhibition function (left panel) and median signal-respond RT as a function of stop-signal delay (SSD; right panel) for the non-right inferior frontal gyrus (non-rIFG) lesion patients after removing stop-signal trials at SSDs of 0, 50, 100, 150, and 250 ms. In the left panel, black bullets show the observed average response rate on stop-signal trials (P(Respond)) for each SSD category, where the SSD categories were defined in terms of the percentiles of the distribution of SSDs for each participant and then averaged across participants. In the right panel, black bullets show the observed average median signal-respond RT (SRRT) for each SSD category, where SSD categories were defined by pooling SSDs over participants before calculating the percentiles. The gray violin plots show the distribution of the 500 average response rates and SRRTs predicted by the model, with the white circles representing the median of the predictions.

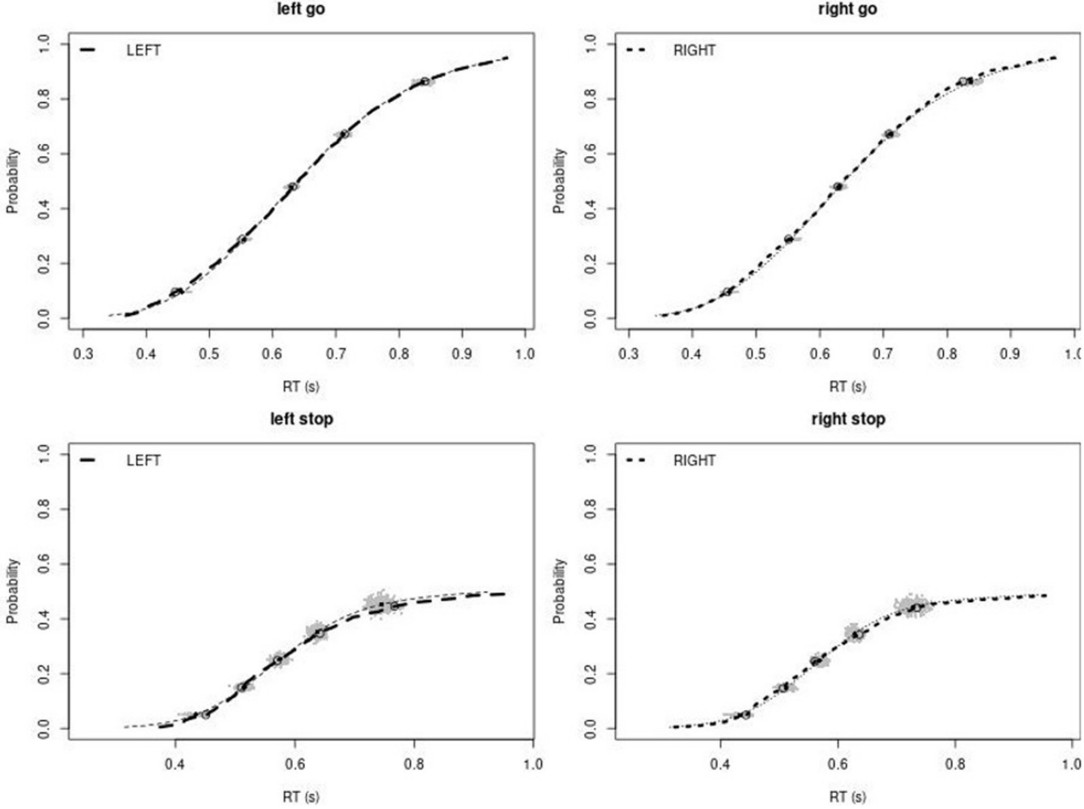

**Appendix 1—figure 11.** Observed and predicted cumulative distribution function (CDF) of Go-RTs (upper panels) and signal-respond RTs (lower panels), separately for left and right stimuli, for the matched comparison group for
*Appendix 1—figure 11 continued on next page*

*Appendix 1—figure 11 continued*

the non-right inferior frontal gyrus (non-rIFG) lesion patients after removing stop-signal trials at SSDs of 0, 50, 100, 150, and 250 ms. The observed and predicted CDFs were averaged across participants. Signal-respond RTs were collapsed across SSD. Thick dashed and dotted lines show the CDF of the observed 'LEFT' and 'RIGHT' responses, respectively. Circles show the 10th, 30th, 50th, 70th, and 90th percentile of the distributions. Thin dashed and dotted lines show the CDF of the predicted 'LEFT' and 'RIGHT' responses, respectively, averaged across the 500 predictions. For each percentile, the gray clouds show the 500 predicted percentiles.

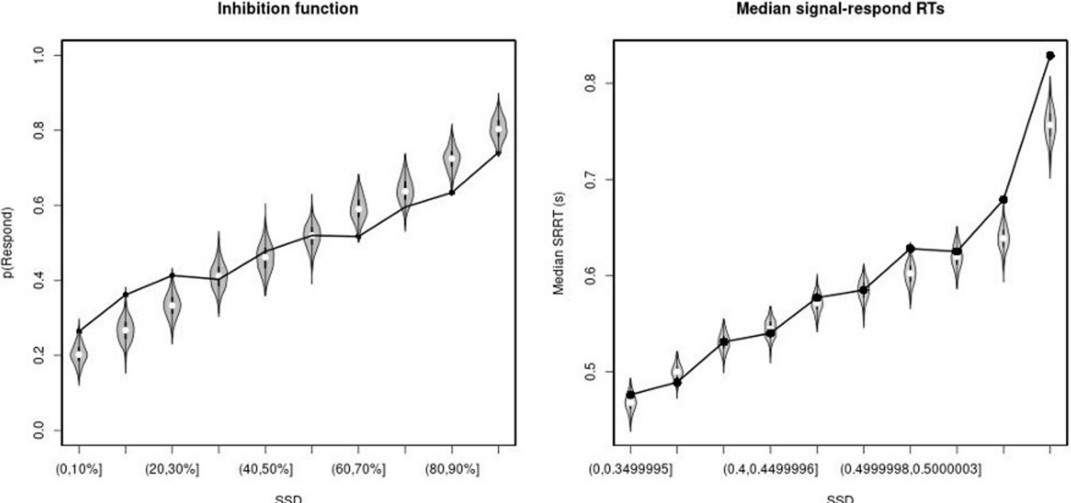

**Appendix 1—figure 12.** Observed and predicted inhibition function (left panel) and median signal-respond RT as a function of stop-signal delay (SSD; right panel) for the matched comparison group for the non-right inferior frontal gyrus (non-rIFG) lesion patients after removing stop-signal trials at SSDs of 0, 50, 100, 150, and 250 ms. In the left panel, black bullets show the observed average response rate on stop-signal trials (P(Respond)) for each SSD category, where the SSD categories were defined in terms of the percentiles of the distribution of SSDs for each participant and then averaged across participants. In the right panel, black bullets show the observed average median signal-respond RT (SRRT) for each SSD category, where SSD categories were defined by pooling SSDs over participants before calculating the percentiles. The gray violin plots show the distribution of the 500 average response rates and SRRTs predicted by the model, with the white circles representing the median of the predictions.

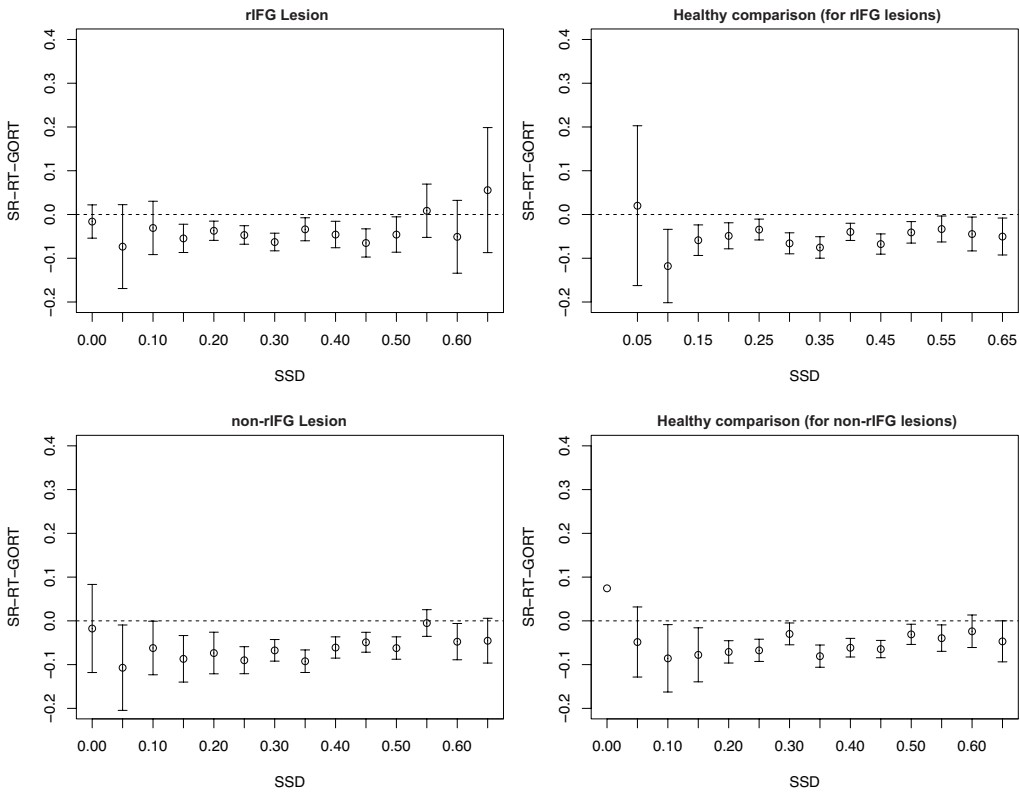

**Appendix 1—figure 13.** Assessment of context independence according to the procedure outlined in *Bissett et al., 2021*. SR-RT, signal-respond RT; SSD, stop-signal delay. None of the groups showed the type of independence violation reported in *Bissett et al., 2021*.

## Non-parametric integration estimates comparisons

We reported the non-parametric integration estimates in the article (*Table 4*) recommended in *Verbruggen et al., 2019*. Below are the statistical comparisons for estimates between groups.

### (1) Non-parametric SSRT estimates comparisons

First, SSRT estimates between lesion vs. its matched healthy comparisons were compared. Paired *t*-test revealed that the rIFG lesion group (*M* = 306.93 ms, SD = 123.77 ms) showed significantly higher SSRT than its matched healthy comparison group (*M* = 230.64 ms, SD = 43.45 ms) (*t*(15) = 2.34, p=0.034, *d* = 0.850). However, there was no significant difference between the non-rIFG lesion group (*M* = 255.47 ms, SD = 46.57 ms) and their matched healthy comparison group (*M* = 238.12 ms, SD = 46.12 ms) (*t*(15) = 1.08, p=0.30, *d* = 0.387). The rIFG lesion group also showed significantly longer SSRT than the healthy comparisons for the non-rIFG lesion group (*t*(30) = 2.08, p=0.046, *d* = 0.761, independent *t*-test). Although SSRT in the rIFG lesion group was numerical longer than SSRT in the non-rIFG lesion group, this difference was not significant (*t*(30) = 1.56, p=0.13, *d* = 0.568, independent *t*-test).

### (2) P(Go error) comparisons

The rIFG lesion group had a significantly increased error rate compared to the other three groups (all *p*s<0.05). The other groups did not significantly differ from each other as the overall go error rates were very low. P(Go error) in the rIFG lesion group was 0.016 while for the other groups it varied between 0.001 and 0.005.

### (3) P(Go miss) comparisons

There was no significant difference between groups in Go miss rates (ps>0.05).

The direct comparison for Stop vs. Go differences in frontal β between two lesion groups showed that the rIFG lesion groups showed significantly reduced β-burst rate difference between Successful-stop and Go trials in the rIFG lesion group (*Appendix 1—figure 14A*, see also 'Results'). Stop vs. Go differences between the rIFG lesion group and the healthy comparison for the non-rIFG lesion group showed that the rIFG lesion group showed numerically reduced β-burst rate difference in Successful-stop and Go trials than that of the healthy comparisons for the non-rIFG lesion group ($t(30) = -1.623$, p=0.115, $d = -0.574$, independent-samples $t$-test). The Failed-stop vs. Go difference showed the same pattern, but it was not significantly different between two groups ($t(30) = -0.757$, p=0.455, $d = -0.268$) (*Appendix 1—figure 14B*).

The two healthy comparison groups did not show significant difference in Successful-stop and Go trials ($t(30) = 1.275$, p=0.212, $d = 0.451$, independent-samples $t$-test), and in between Failed-stop and Go trials ($t(30) = .673$, p=0.506, $d = 0.238$, independent-samples $t$-test) (*Appendix 1—figure 14C*).

Finally, Stop vs. Go differences between the non-rIFG lesion group and the healthy comparison for the rIFG lesion group showed no significant difference in Successful-stop and Go trials ($t(30) = 0.786$, p=0.438, $d = -0.278$, independent-samples $t$-test). There was no significant difference in Failed-stop vs. Go difference between the non-rIFG lesion group and the healthy comparisons for the rIFG lesion group ($t(30) = 0.397$, p=0.695, $d = 0.140$, independent-samples $t$-test) (*Appendix 1—figure 14D*).

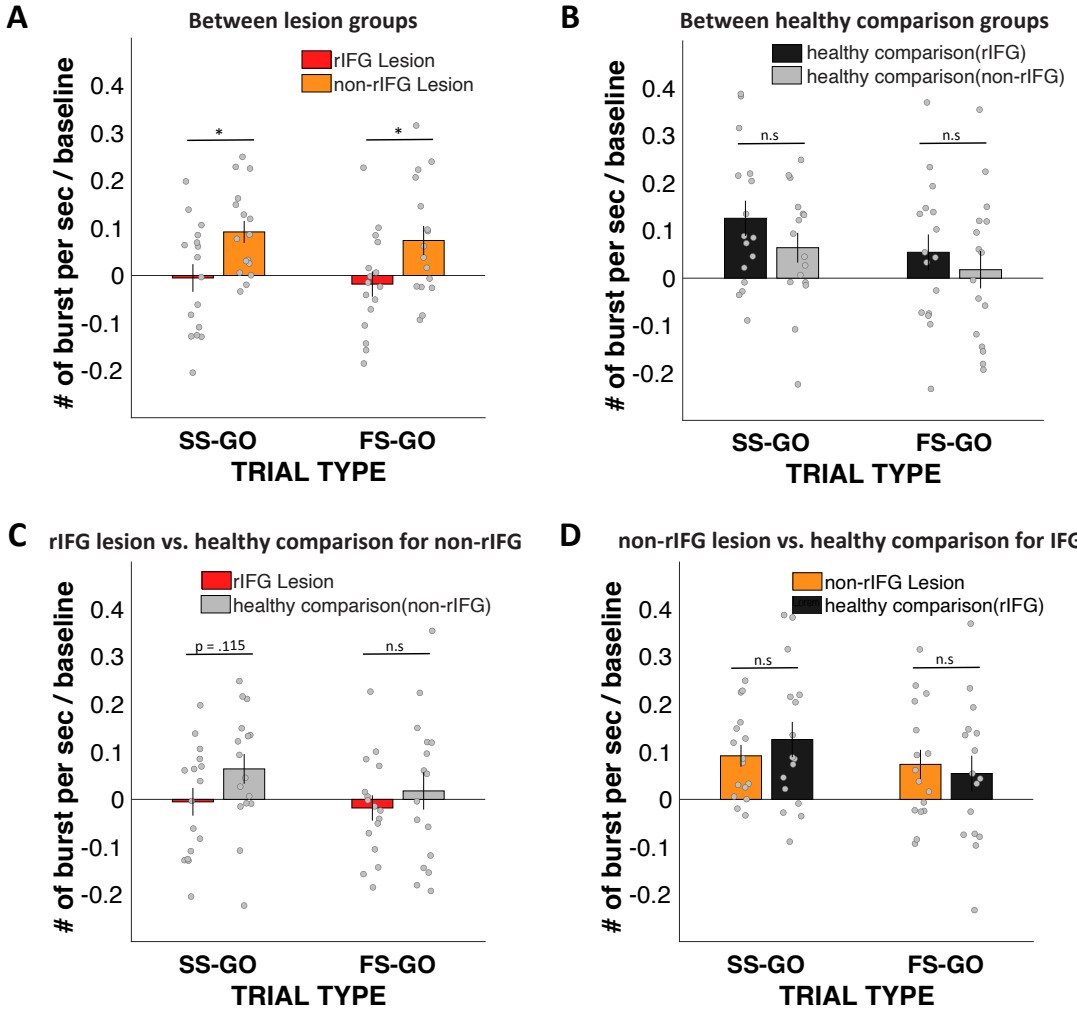

**Appendix 1—figure 14.** Stop vs. Go difference (Successful-stop – Go and Failed-stop – Go) in frontal β-burst rates between groups. (**A**) Between two lesion groups: right inferior frontal gyrus (rIFG) lesion group vs. non-rIFG lesion group. (**B**) rIFG lesion group vs. healthy comparison for the non-rIFG lesion group. (**C**) Between two healthy comparison groups: healthy comparison for the rIFG lesion group vs. healthy comparison for the non-rIFG lesion group. (**D**) Non-rIFG lesion group vs. healthy comparison for the rIFG lesion group.

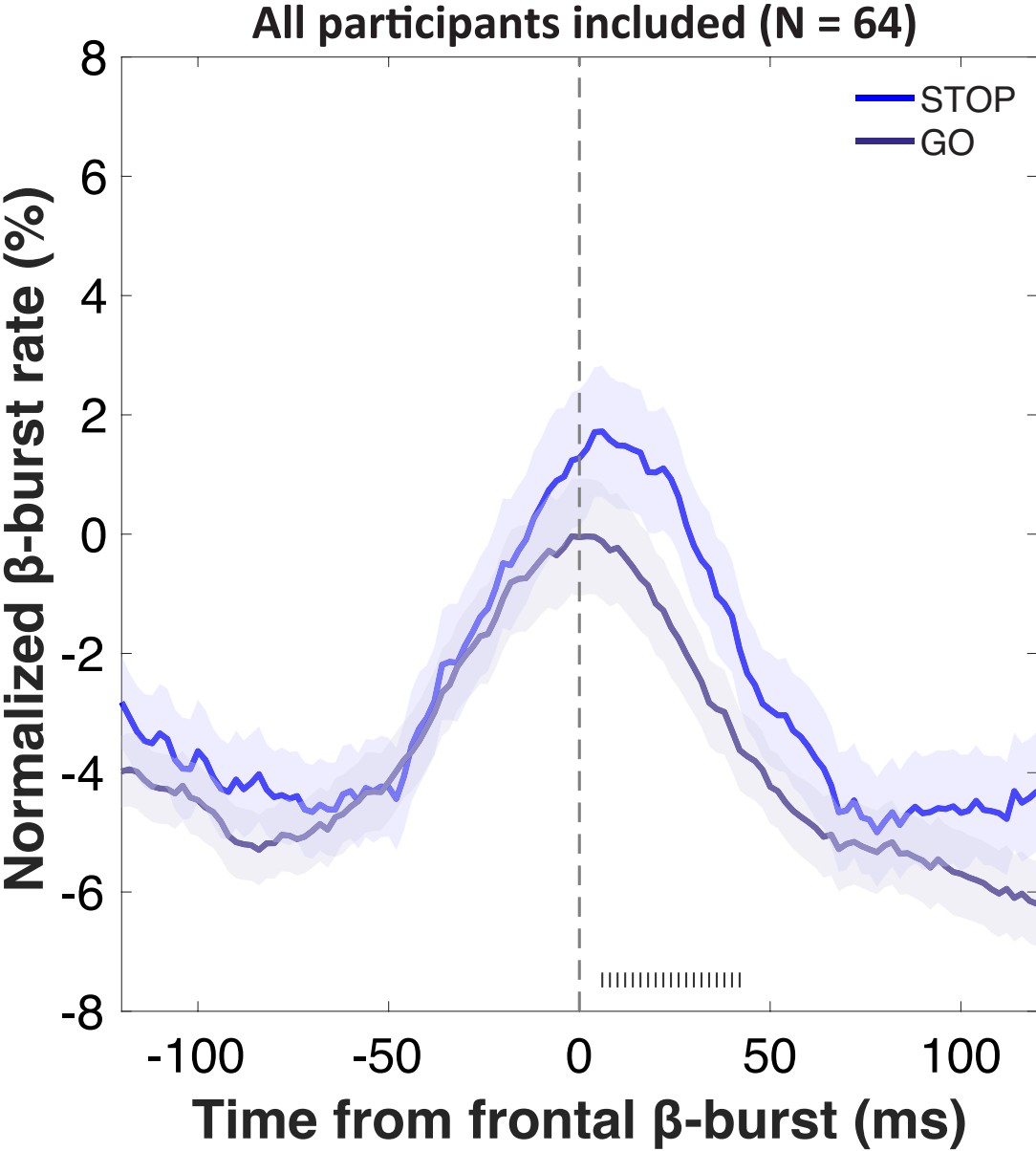

**Appendix 1—figure 15.** Increases in sensorimotor β-bursts following frontal β-bursts on stop- and go-trials. Gray hashes on the bottom denote significant differences between STOP and GO trials at p<0.05.

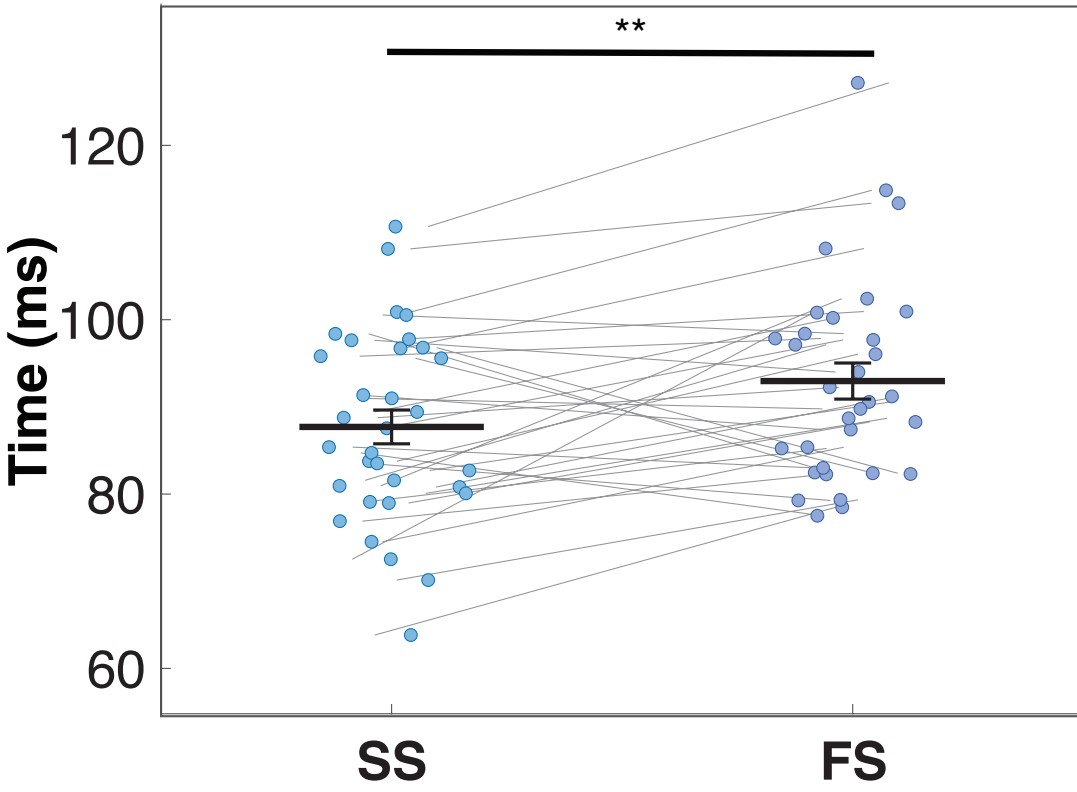

**Appendix 1—figure 16.** Timing of first beta burst after stop-signal. Frontal β-bursts in healthy comparisons occur earlier on successful (SS) compared to failed (FS) stop-trials. This difference was significant: $t(31) = -2.93$, p=0.003, $d = -0.518$, one-sided.

