## [Editor Report]

This study takes a fresh view of the hypothesis that right inferior frontal gyrus is critical in inhibitory control in humans, as assessed by the widely used stop-signal task. It applies recent development in modeling and EEG measures in patients with focal brain damage, yielding causal insights. The findings are important and the evidence is convincing.

---

## [Decision Letter]

**Decision letter after peer review:**

Thank you for submitting your article "Right inferior frontal cortex damage impairs the initiation of inhibitory control, but not its implementation" for consideration by *eLife*. Your article has been reviewed by 2 peer reviewers, and the evaluation has been overseen by David Badre as the Reviewing Editor and Michael Frank as the Senior Editor. The following individual involved in the review of your submission has agreed to reveal their identity: Lesley K Fellows (Reviewer #1).

The reviewers found this to be a strong, hypothesis-driven study that makes fresh progress on an important topic. They were in consensus that it may be suitable for *eLife* with appropriate revision. They did agree on several points that should be addressed. They focus on three broad issues related to the characterization of the lesions, a fuller picture of the data, and some points of clarity in the discussion. The reviewers have discussed their reviews with one another, and the Reviewing Editor has drafted this summary to help you prepare a revised submission.

Essential revisions:

As this is a lesion study, it is crucial to take care in the interpretation of the lesion. There were several points to consider in this regard:

(1) The paper sets up as a follow-up to the Aron et al. (2003) paper, but it was not fully clear whether the location of the present lesion if rIFC is in the same region of IFC that was the region of interest of focus in Aron. Indeed, to some extent looks somewhat rostral to that lesion, but it is difficult to tell with certainty.

(2) The term right inferior frontal cortex is used throughout regarding the region of interest. However, the lesions under study here are not solely cortical. I think it would be preferable to describe this as right inferior frontal gyrus, or right ventrolateral frontal lobe. Or at least discuss this issue, if the authors prefer to stick with the current terminology.

(3) The lesion overlap is as extensive in the insula as in the right ventral frontal lobe (not surprising, as damage tends to covary in these regions). This warrants comment, particularly given that insula is also frequently implicated in fMRI studies of the stop-signal task.

(4) Reviewers may have missed it, but the causes of the lesions should be provided. Were patients taking psychoactive medications?

(5) Table 1 should include an indication of variance for chronicity, such as IQR or range. Some comment seems warranted on the seemingly very large difference in chronicity between the IFC and lesioned control groups. The IFC lesions seem to be very chronic indeed; were some of these lesions in adolescence or early adulthood? Effects at those developmental stages might be quite different than acquired brain injury in middle- or old-age, with implications for interpretation of the main results.

Reviewers noted a number of places where the analyses were unclear or they were unsure how dependent the results are on the particular abstract analysis being conducted; though preregistration was noted as a strength. They had several requests for analysis to get a fuller picture of the data. I provide the specific comments along these lines here:

(6) Figure 3 is a challenge to follow. I'm not sure about showing key findings 'normalized' to baseline, what is the justification for showing a ratio rather than the raw data? The statistical test approach for the β burst result is not easy to follow. If I understand correctly, a difference between the rIFC group and their control group is present, while no difference is detected between the other lesion group and (different) controls. This is pretty indirect evidence for lesion location specificity; the mean β burst rate seems quite (numerically) different in the non-frontal lesion-control group compared to all other groups, and there is generally a lot of variability. The second analysis involving subtraction of normalized β burst rates seemed potentially unjustified arithmetic. It would be great to see something a little closer to the raw data to get a better sense of this part of the results. Were all of these analyses pre-registered?

(7) I was struggling with the very strong focus on TF basically being the only relevant behavioral observation here. A five-fold increase is certainly a lot, but it is also a five-fold increase of a very low number. Additionally, a residual SSRT effect of 35 ms, in fact, is also still a pretty large effect with regard to earlier group-difference observations. This residual effect is put into perspective some (concerning mu vs. tau), but rather in passing, as also being more in line with an attentional account, but this was hard to follow, and I am not sure how legitimate this is. I would down-tone this exclusivity a bit, and/or explore the residual SSRT effect some more (see below). Since I would also recommend providing the non-parametric SSRT estimate for reference (see below), I also think that a better metric might be how much of the SSRT group effect is explained by TF vs. what remains after accounting for it (say that the integration-approach SSRT would be 300 for the rIFG group and 240 for the control group, then accounting for TF would reduce this difference by approximately 30 ms, and still leave a residual difference of again approximately 30 ms). Maybe this is also not a perfect metric, but I think it would be significantly more intuitive, as far as the respective contributions of TF vs. residual SSRT go.

(8) Further, related to the preceding point, I noticed that the TF rate for all other groups is also really low in an absolute sense. Do the authors share the opinion that e.g. the 2% TF rate for the non-rIFC patients is surprisingly low, given what has usually been found in the literature? If yes, what might be the reason?

(9) As already stated, I would find it useful to also provide SSRT estimates from the hitherto pretty ubiquitous non-parametric approach for reference? Also, it would be reassuring to see that the assumptions of that approach are met, just to get another viewing angle on data quality. Relatedly, I had the impression that some information that is recommended to be provided in the cited consensus paper (e.g., average SSDs, p(inhibit), SRRT, etc.) is not available. Please add this. This is also true for Go misses, which I realize likely also contain some slow responses, but it would nonetheless be important to see this data. If need be, the data of the groups that do not have the time-out issue could also be limited to a 1000-ms deadline, in order to match this feature.

(10) Sometimes the patient groups are only compared to their respective control group, and sometimes directly with each other. I am wondering what the rationale is to sometimes also do the latter, and sometimes not. Again, maybe a more complete set of analyses (a lot of which could go to the supplementary material) would likely be good.

(11) Maybe you could discuss a bit more that the rIFG group has no condition effect whatsoever for mid-frontal β bursts in the sense that there are pretty exactly as many in successful and failed inhibition trials as in Go trials. I assume that this is still relevant from an inferential standpoint, and could furthermore be further explored empirically, e.g. in the sense of doing an analysis akin to what is presented in Figure 4 for Go-trials, in which there happened to be a (early) β burst. Should/does this then also lead to an increased likelihood of a β burst over the motor cortex?

(12) Related to this, I was expecting a similar analysis also for failed stop trials. Would such an analysis not also potentially be relevant (e.g., motor-cortex β bursts might be triggered, but too late, or not triggered at all, etc.). I appreciate that this dataset is not of the size and quality of a well-powered study in healthy participants, but this could probably be explored (again, no problem to do so in the supplementary material).

(13) I was unclear about how the manuscript dealt with the misfit in the supplementary material. What was the rationale for excluding the short SSDs and how does the improved fit after exclusion speak to the validity of the findings? I did read the respective section in the method section, but it was not sufficiently clear to me. I would certainly appreciate some more explanation.

(14) Related to the fact that short SSDs are found to be an issue here, and the recent publication by Bissett et al. (which is also cited here), highlighting the possibly important confounding role of short SSDs via context-independence violations, I think it would be good to systematically test for such violations, as prescribed in their paper. Again, pretty certainly just material for the supplementary material!

Reviewers also highlighted several places where the text could be clarified, its contact with the literature broadened, and its underlying assumptions more explicitly acknowledged.

(15) The paper addresses only the influential Aron et al. lesion study. However, there have been other lesion studies examining frontal contributions to stop-signal task performance and more addressing inhibitory control more generally, i.e. using go-no go tasks, for example (Swick et al. 2008; Picton et al. 2007). Even before considering the issue raised in the present study, the lesion evidence for a specific role for right IFG in inhibitory control was weak (Floden & Stuss, 2006; Yeung et al., JOCN, 2021). More generally, the positioning of this study made me feel I was being dropped into a very 'insider club' issue-a somewhat broader approach to both the introduction and discussion would be helpful in situating this study beyond the research groups that are deeply engaged in stop-signal task research.

(16) I found the frequently used concept of "implementation of inhibitory control" a bit difficult and would appreciate some further elaboration. I suppose there could also be a view in which the triggering of inhibition is in fact basically already its implementation, with everything downstream following a ballistic/stereotypical course. As such, I would be extremely curious whether the authors could somehow pinpoint what the remaining SSRT effect (or rather variance thereof) is related to. If the variance in the residual SSRT would be related to differential timing of the triggering of the stop process (via β bursts in the IFG), it would feel more (to me) like the right IFG still being a key structure for response inhibition. If in turn, it is a simple on/off switch, with no relevance beyond whether it is triggered or not (i.e., no relationship to the residual SSRT, which would be explained by other/later processes), I would agree yet more with the need for a fundamental reinterpretation of its role.

(17) Figure 2A seems to have garbled text; I was not clear about the proportion of stop signal trials or whatever other design features might or might not make this version of the task comparable to the existing literature. I guess this must have been explained in that figure before it was garbled, perhaps in pdf-production? It might be worth explaining how the task is 'optimized' for this particular modeling approach in the text-concretely, what is different from widely-used versions of the SST?

(18) Table 2 might be easier to appreciate (and relate to the literature) if it was divided into the conventional SST outcomes measures and the hierarchical model parameters, rather than mixing them. Did the groups differ on error rate?

Finally, the reviewers offered some minor points that you may choose to consider in your revision.

– I would find it helpful if the authors could overlay the borders of the different IFG subregions, as well as of the anterior insula in Figure 1A. I appreciate that spatial specificity is limited here, but given the numerous indications of dissociations between these areas, I would still appreciate easily being able to see this in reference to the overlap map of the lesions.

– Just out of curiosity: is chronicity (Tab. 1) possibly relevant? I don't really think so, but the difference between the patient groups is very pronounced.

– Maybe I am mistaken, but I didn't see any stats underlying the finding illustrated in Figure 4A. Also, I would find it relevant to know how many trials have gone into these various (elegant) analyses.

– In the Discussion section, would an explicit reference to the pause-then-cancel model not be relevant?

---

## [Author Response]

Essential revisions:As this is a lesion study, it is crucial to take care in the interpretation of the lesion. There were several points to consider in this regard:(1) The paper sets up as a follow-up to the Aron et al. (2003) paper, but it was not fully clear whether the location of the present lesion if rIFC is in the same region of IFC that was the region of interest of focus in Aron. Indeed, to some extent looks somewhat rostral to that lesion, but it is difficult to tell with certainty.

We appreciate the question. We have added anatomical labels to the lesion overlap figure (Figure 1) in the revised version of the manuscript.

It is correct that the lesion is slightly more rostral than the Aron et al. lesion, but both centers of gravity were predominantly located in pars triangularis of the right inferior gyrus. Within the constraints of lesion experimentation, we believe that this is about as close of an overlap as can be expected by a replication in an independent sample. We hope the reviewers agree. We have included the new figure (which also includes labels for the anterior insula to address a later point; see above) in the revised version of the manuscript.

(2) The term right inferior frontal cortex is used throughout regarding the region of interest. However, the lesions under study here are not solely cortical. I think it would be preferable to describe this as right inferior frontal gyrus, or right ventrolateral frontal lobe. Or at least discuss this issue, if the authors prefer to stick with the current terminology.

We agree. We have changed the nomenclature to “right inferior frontal gyrus” throughout.

(3) The lesion overlap is as extensive in the insula as in the right ventral frontal lobe (not surprising, as damage tends to covary in these regions). This warrants comment, particularly given that insula is also frequently implicated in fMRI studies of the stop-signal task.

Indeed. As in most (or all) lesion studies of the ventrolateral frontal lobe, there was some damage to the anterior insula.

With regard to the insula’s role in stopping, previous work has shown that the anterior insula seems to be primarily activated on unsuccessful stop trials (Cai et al., J Neuro 2014), in line with its purported role in error processing (Ullsperger et al., Brain Structure and Function 2010). Moreover, intracranial recordings have shown that the insula tends to be active after rIFG, and indeed after SSRT (i.e., probably does not contribute to stopping itself, Bartoli et al., HBM 2018), which also makes sense in this context. It is hence unlikely to account for the findings in the current study. We discuss these implications in the revised version of the manuscript on p. 22.

(4) Reviewers may have missed it, but the causes of the lesions should be provided. Were patients taking psychoactive medications?

Agreed. We have added the following statement about lesion etiology into the revised version of our manuscript on p. 22: “Lesion etiologies included: ischemic stroke (n=13), hemorrhagic stroke (n=4), focal contusion (n=2), AVM or cavernoma resection (n=5), benign tumor resection (n=3), herpes simplex encephalitis (n=2), cyst resection (n=1), abscess resection (n=1), and epilepsy resection (n=1).”

Finally, some patients in the database were indeed taking psychoactive medication, as is to be expected given their etiology and age. However, patients were excluded from participation if they were taking medication at dosages that can cause cognitive side effects. We have included a statement to that effect into the revised version of the manuscript on p. 22.

(5) Table 1 should include an indication of variance for chronicity, such as IQR or range. Some comment seems warranted on the seemingly very large difference in chronicity between the IFC and lesioned control groups. The IFC lesions seem to be very chronic indeed; were some of these lesions in adolescence or early adulthood? Effects at those developmental stages might be quite different than acquired brain injury in middle- or old-age, with implications for interpretation of the main results.

Indeed. The variance in chronicity is a side effect of our attempts to achieve sufficient statistical power (i.e., sample size) while maintaining a very constrained lesion location. IQR information for chronicity was included in the revised table 1. Moreover, there was a calculation mistake in our initial chronicity estimates that has been corrected, and we apologize for the oversight (the original pattern of longer chronicity in the lesion group remains in place, though). Indeed, three individuals in the rIFG group did have developmental onset lesions. We have repeated the main analyses of interest with these participants removed and have found that both the trigger failure parameter differences and the β burst differences remained highly significant.

rIFG vs. matched healthy comparison N = 13

Trigger failure parameter, probit scale, mean(SEM)

-1.3498 (0.26447) vs. -3.5261 (0.19955); t(12) = 5.5029, p * = 0.00013562, d = 1.5262 (For simplicity’s sake, this analysis was performed using standard frequentist t-testing on the individual model parameters from the BEESTS model that was fit to all participants’ data (i.e., the one reported in the main manuscript). We acknowledge that using point estimates of participant-level parameters from hierarchical Bayesian models in frequentist tests can inflate Type-I error rates as a result of hierarchical shrinkage. Due to the relatively low sample size, and hence modest shrinkage, this effect is expected to be relatively small for the present analysis. We would be willing to re-fit the model using only these participants and calculate exact Bayesian p-values (as was done for the full analysis in the main manuscript), but given the quite sizable group differences in the parameter estimates and since this is merely a control analysis, we held off on doing so at this point, as the full model would take a very long time to run.)

β burst rates for SS-GO at FCz, mean(SEM) with normalization

-0.019724 (0.033792) vs. 0.1296 (0.044518); t(12) = -4.0767, p * = 0.0015351, d = -1.1307.

This shows that our results do not seem to be attributable the presence of developmental onset lesions or the differences in chronicity (which were negligible after the removal of the three subjects in the above analysis). We made mention of these analyses in the revised version of the manuscript on p. 23.

Reviewers noted a number of places where the analyses were unclear or they were unsure how dependent the results are on the particular abstract analysis being conducted; though preregistration was noted as a strength. They had several requests for analysis to get a fuller picture of the data. I provide the specific comments along these lines here:(6) Figure 3 is a challenge to follow. I'm not sure about showing key findings 'normalized' to baseline, what is the justification for showing a ratio rather than the raw data? The statistical test approach for the β burst result is not easy to follow. If I understand correctly, a difference between the rIFC group and their control group is present, while no difference is detected between the other lesion group and (different) controls. This is pretty indirect evidence for lesion location specificity; the mean β burst rate seems quite (numerically) different in the non-frontal lesion-control group compared to all other groups, and there is generally a lot of variability. The second analysis involving subtraction of normalized β burst rates seemed potentially unjustified arithmetic. It would be great to see something a little closer to the raw data to get a better sense of this part of the results. Were all of these analyses pre-registered?

To answer the last question first: the β burst analyses were not pre-registered, only the behavioral/computational modeling analyses and hypotheses were, as indicated by the statement in the manuscript. That is simply because the pre-registration and beginning of data collection predate the ‘discovery’ of the importance of scalp-recorded β bursts for motor inhibition in Wessel (J Neuro 2020) and Jana and colleagues (*eLife* 2020).

With regards to the β burst metrics themselves: they were directly adapted from the latter paper (Jana et al., *eLife* 2020), including the exact normalization procedure. For the sake of completion, here are the results of the non-normalized burst count analysis:

First, and most importantly, the pattern of significance remains exactly the same. Second, the basic rate of β bursts does not appear to differ dramatically between the groups. We have included a mention of these additional analyses (and the rationale for the normalization procedure) in the revised version of the manuscript on p. 31.

Finally, with regards to the question of lesion specificity: The reviewers may have missed that we did actually directly test whether there was a difference in stop-related β bursts between the rIFG and non-rIFG lesion groups (cf., p. 13 of the revised manuscript). That test revealed a significant difference.

(7) I was struggling with the very strong focus on TF basically being the only relevant behavioral observation here. A five-fold increase is certainly a lot, but it is also a five-fold increase of a very low number. Additionally, a residual SSRT effect of 35 ms, in fact, is also still a pretty large effect with regard to earlier group-difference observations. This residual effect is put into perspective some (concerning mu vs. tau), but rather in passing, as also being more in line with an attentional account, but this was hard to follow, and I am not sure how legitimate this is. I would down-tone this exclusivity a bit, and/or explore the residual SSRT effect some more (see below). Since I would also recommend providing the non-parametric SSRT estimate for reference (see below), I also think that a better metric might be how much of the SSRT group effect is explained by TF vs. what remains after accounting for it (say that the integration-approach SSRT would be 300 for the rIFG group and 240 for the control group, then accounting for TF would reduce this difference by approximately 30 ms, and still leave a residual difference of again approximately 30 ms). Maybe this is also not a perfect metric, but I think it would be significantly more intuitive, as far as the respective contributions of TF vs. residual SSRT go.

The suggested approach is intuitively appealing, but has the potential to confound genuine differences in SSRTs resulting from the presence of trigger failures with differences resulting from using different estimation methods (i.e., non-parametric individual level vs. parametric hierarchical) that rely on different assumptions. Moreover, this approach assumes that P(TF) reflects attention and SSRT purely reflects inhibition. The BEESTS model subscribes to the former, but not to the latter: in line with the standard race model, BEESTS assumes that SSRT captures the duration of a chain of processes that contribute to stopping, such as attentional/perceptual, decisional, and (inhibitory) motor-related processes (e.g., Logan et al. 2014; for an overview, see Matzke et al., 2018). Our aim with the model-based analysis of the stopping data is to disentangle the relative contribution of these processes using a fine-grained analysis of SSRT distributions. In particular, the reason we argue for a mostly attentional deficit is not because the SSRT difference was small after accounting for trigger failures, but because of the particular pattern of parameter differences where P(TF) and ***μ***_stop_ were larger in the rIFG lesion group than in the matched comparison group whereas ***τ***_stop_ was the same. Matzke et al. (2017) argued, based on earlier results in the RT modeling literature, that this pattern in SSRT distributions is consistent with differences in the speed of encoding the stop signal and not with differences in the decisional/inhibitory component of SSRT. As we now point out in the revised manuscript, this is the reasoning we relied on in the present work.

(8) Further, related to the preceding point, I noticed that the TF rate for all other groups is also really low in an absolute sense. Do the authors share the opinion that e.g. the 2% TF rate for the non-rIFC patients is surprisingly low, given what has usually been found in the literature? If yes, what might be the reason?

Our trigger failure estimates are indeed on the low side, but the 95% credible intervals in the two control groups overlap with other studies with similar easy choice-based SSTs. For instance, Matzke et al. 2017 (APP) reported TF credible intervals for control participants ranging between [0.04,0.12] for data originally reported in Hughes et al. (2012) and [0.06,0.16] for data originally reported in Badcock et al. (2002). Our credible intervals in the current study were [0,0.12] for the rIFC comparisons and [0.02,0.08] for the non-rIFC comparisons. The only papers that reported substantially higher rates of trigger failures were by Skippen and colleagues, who used a very unconventional task that involved a complicated go-choice and was specifically designed to induce fatigue (and hence, increase trigger failure probability).

(9) As already stated, I would find it useful to also provide SSRT estimates from the hitherto pretty ubiquitous non-parametric approach for reference? Also, it would be reassuring to see that the assumptions of that approach are met, just to get another viewing angle on data quality. Relatedly, I had the impression that some information that is recommended to be provided in the cited consensus paper (e.g., average SSDs, p(inhibit), SRRT, etc.) is not available. Please add this. This is also true for Go misses, which I realize likely also contain some slow responses, but it would nonetheless be important to see this data. If need be, the data of the groups that do not have the time-out issue could also be limited to a 1000-ms deadline, in order to match this feature.

We agree. We have added this information in the revised version of the manuscript in Table 4 and in Appendix 1.

(10) Sometimes the patient groups are only compared to their respective control group, and sometimes directly with each other. I am wondering what the rationale is to sometimes also do the latter, and sometimes not. Again, maybe a more complete set of analyses (a lot of which could go to the supplementary material) would likely be good.

As far as we know, the only analysis that was not run in the initial version of the paper was the direct comparison between the BEESTS parameters of the rIFG lesion group vs. non-rIFG lesion group. That analysis has now been included in the revised version of the manuscript (Table 3). The results are very similar to the primary result of rIFG lesion group vs. matched healthy comparison group (Table 2). The rest of β burst analysis for Stop vs. Go difference between group-comparisons were reported in Appendix figure 14. Thank you for pointing this out.

(11) Maybe you could discuss a bit more that the rIFG group has no condition effect whatsoever for mid-frontal β bursts in the sense that there are pretty exactly as many in successful and failed inhibition trials as in Go trials. I assume that this is still relevant from an inferential standpoint, and could furthermore be further explored empirically, e.g. in the sense of doing an analysis akin to what is presented in Figure 4 for Go-trials, in which there happened to be a (early) β burst. Should/does this then also lead to an increased likelihood of a β burst over the motor cortex?

We now make mention of the fact that there was no increase of β bursts at all in the rIFG lesion group (p. 17), though we want to be cautious about such statements, as they technically imply the confirmation of the null hypothesis in a (relatively) small sample. Moreover, if we understand the question correctly, the reviewer is wondering whether β bursts on go-trials are also followed by increase sensorimotor β bursts. That was generally the case, though to a lesser extent than for STOP trials:

This analysis was added to Appendix 1 for the revised manuscript.

(12) Related to this, I was expecting a similar analysis also for failed stop trials. Would such an analysis not also potentially be relevant (e.g., motor-cortex β bursts might be triggered, but too late, or not triggered at all, etc.). I appreciate that this dataset is not of the size and quality of a well-powered study in healthy participants, but this could probably be explored (again, no problem to do so in the supplementary material).

As for the go-trials, frontal β bursts on failed stop-trials also lead to an increase in sensorimotor β bursting. However, as the reviewer correctly surmises, the main difference between failed stop-trials and successful stop-trials is in the timing of the frontal β bursts, which occurs later on failed vs. successful stop trials (Jana et al., *eLife* 2020), leading to reduced β burst counts prior to SSRT (Wessel, JNeuro 2020). The same was true for our sample: t(31) = -2.93, p * = 0.003, d = -.518.

This analysis has been added to Appendix 1 as well.

(13) I was unclear about how the manuscript dealt with the misfit in the supplementary material. What was the rationale for excluding the short SSDs and how does the improved fit after exclusion speak to the validity of the findings? I did read the respective section in the method section, but it was not sufficiently clear to me. I would certainly appreciate some more explanation.

As we explained in the main text and illustrated in the Appendix 1, the BEESTS model showed a quantitatively small misfit (i.e., underprediction) to the average inhibition function at short SSDs in the non-rIFG lesion patients and their matched comparisons (see Appendix figure 6 and 8). To examine the robustness of the results to a possible model misspecification, we sequentially removed all stop-signal trials from the data of the non-rIFG lesion and the matched comparison group at SSDs of 0, 50, 100, 150, and 250 ms (i.e., SSDs where the misfit occurred), re-fit the model, and re-assessed the model’s descriptive accuracy. As shown in Appendix 1, descriptive accuracy improved as stop-signal trials on short SSDs were removed (Appendix figure 10 and 12): after removing stop trials at SSDs with 0 to 250 ms, the model was able to accurately account for the observed inhibition function in both groups. Importantly, qualitative conclusions about group differences were the same whether or not stop trials with short SSDs were included in the analysis. This demonstrates the robustness of the results and the validity of our conclusions. We now explain this more clearly in the revision.

(14) Related to the fact that short SSDs are found to be an issue here, and the recent publication by Bissett et al. (which is also cited here), highlighting the possibly important confounding role of short SSDs via context-independence violations, I think it would be good to systematically test for such violations, as prescribed in their paper. Again, pretty certainly just material for the supplementary material!

We included the test of context independent violations proposed by Bissett et al. (2021) in the Appendix 1 (Appendix figure 13). There was no evidence of the type of independence violation reported by Bissett et al. (2021); Bissett et al. found that violations were manifest in the signal-respond RTs on short SSDs, whereas the misfit in our data was restricted to response rates. We would like to emphasize again that this misfit was quantitatively small and did not influence our conclusions.

Reviewers also highlighted several places where the text could be clarified, its contact with the literature broadened, and its underlying assumptions more explicitly acknowledged.(15) The paper addresses only the influential Aron et al. lesion study. However, there have been other lesion studies examining frontal contributions to stop-signal task performance and more addressing inhibitory control more generally, i.e. using go-no go tasks, for example (Swick et al. 2008; Picton et al. 2007). Even before considering the issue raised in the present study, the lesion evidence for a specific role for right IFG in inhibitory control was weak (Floden & Stuss, 2006; Yeung et al., JOCN, 2021). More generally, the positioning of this study made me feel I was being dropped into a very 'insider club' issue-a somewhat broader approach to both the introduction and discussion would be helpful in situating this study beyond the research groups that are deeply engaged in stop-signal task research.

We agree. The downside of having a pre-registered study with a very circumscribed approach and a clear replication target is that one runs the risk of being too narrow in scope in framing the work. We re-written the introduction of the paper to hopefully do the wider literature more justice (p. 3-4).

(16) I found the frequently used concept of "implementation of inhibitory control" a bit difficult and would appreciate some further elaboration. I suppose there could also be a view in which the triggering of inhibition is in fact basically already its implementation, with everything downstream following a ballistic/stereotypical course. As such, I would be extremely curious whether the authors could somehow pinpoint what the remaining SSRT effect (or rather variance thereof) is related to. If the variance in the residual SSRT would be related to differential timing of the triggering of the stop process (via β bursts in the IFG), it would feel more (to me) like the right IFG still being a key structure for response inhibition. If in turn, it is a simple on/off switch, with no relevance beyond whether it is triggered or not (i.e., no relationship to the residual SSRT, which would be explained by other/later processes), I would agree yet more with the need for a fundamental reinterpretation of its role.

To us, the residual variance in SSRT is likely due to the processing speed in the fiber tracts downstream from rIFC, including in the basal ganglia. The integrity of white matter connections from rIFC to STN predict stopping speed (Coxon et al., J Neuro 2012), which has recently been confirmed using intracranial direct electrical stimulation as well (Chen et al., Neuron 2020). We have expanded on this thought in the revised version of the discussion on p. 21.

(17) Figure 2A seems to have garbled text; I was not clear about the proportion of stop signal trials or whatever other design features might or might not make this version of the task comparable to the existing literature. I guess this must have been explained in that figure before it was garbled, perhaps in pdf-production? It might be worth explaining how the task is 'optimized' for this particular modeling approach in the text-concretely, what is different from widely-used versions of the SST?

We apologize for the oversight with regards to the figure, the PDF conversion did not go as planned and this was not detected prior to submission. We have fixed this figure in the revised version of the manuscript.

In terms of ‘optimizing’ the task for the purpose of BEESTS: this solely meant a higher trial count than perhaps typical, as well as an ‘open’ response window that detected responses made even after the deadline (so as to not truncate the RT distribution). Other than that, the task is identical to the typical versions in the literature. This has been explicitly pointed out in the revised version of the manuscript.

(18) Table 2 might be easier to appreciate (and relate to the literature) if it was divided into the conventional SST outcomes measures and the hierarchical model parameters, rather than mixing them. Did the groups differ on error rate?

We apologize for the misunderstanding – Table 2 only includes the model parameters from the BEESTS analysis (in the revision, we have now added the standard measurements in Table 4 as well). Of course, there is some overlap between the naming of variables. However, we hope that our presentation in two separate tables in the revised manuscript avoids this confusion.

In terms of error rate, the rIFG lesion group had a significantly increased error rate compared to all other three groups (*ps* <.05), but the overall rates were very low (1.62% in the rIFG lesion group,.49% in the non-rIFG lesion group,.14% in healthy comparison for rIFG lesion and.43% in the healthy comparison for the non-rIFG lesion group).

Finally, the reviewers offered some minor points that you may choose to consider in your revision.– I would find it helpful if the authors could overlay the borders of the different IFG subregions, as well as of the anterior insula in Figure 1A. I appreciate that spatial specificity is limited here, but given the numerous indications of dissociations between these areas, I would still appreciate easily being able to see this in reference to the overlap map of the lesions.

See above.

– Just out of curiosity: is chronicity (Tab. 1) possibly relevant? I don't really think so, but the difference between the patient groups is very pronounced.

See above.

– Maybe I am mistaken, but I didn't see any stats underlying the finding illustrated in Figure 4A. Also, I would find it relevant to know how many trials have gone into these various (elegant) analyses.

Trial numbers for Figure 4A were as follows:

Trials with bursts: M = 60.734, SD = 10.310

Trials without bursts: M = 14.797, SD = 7.458

However, the more relevant counts are for the between-groups comparisons (Figure 4B), as Figure 4A is merely a replication of prior work.

rIFG lesion: M = 57.688, SD = 14.263

matched healthy for rIFG lesion: M = 63.50, SD = 7.118

non-rIFG lesion: M = 61.50, SD = 7.729

matched healthy for non-rIFG lesion: M = 60.25, SD = 10.662

There was no significant difference in trials included in group comparison.

We have made mention of these trial counts in the revised version of the manuscript on p. 16.

– In the Discussion section, would an explicit reference to the pause-then-cancel model not be relevant?

Indeed. We have made reference to this work in the revised version of the discussion.